# Fishing during extreme heatwaves alters ecological interactions and increases indirect fishing mortality in a ubiquitous nearshore system
Jeff C. Clements [1,2] ✉, Sarah Harrison[1,2], Mylène Roussel[1], Jillian Hunt[1], Brooke-Lyn Power[1,2] & Rémi Sonier[1]

Heatwaves may have multifaceted ecological impacts; however, field studies assessing the ecological ramifications of nearshore fishing during heatwaves are rare. We leverage a field experiment simulating clam fishing to document such effects on a ubiquitous ecological system at the land-sea interface. During monthly field trials from May-September 2024, we experimentally fished clams at low tide and tracked reburrowing and mortality rates of marked, sub-legal sized clams returned to the sediment. Half of the clams were protected from crab predation and estimates of predator and scavenger activity were recorded. Clams typically reburrowed quickly and mortality was low. During the heatwave, however, clams appeared unhealthy, failed to reburrow, and suffered near-complete mortality. Predator activity in experimental plots was >4× higher during the heatwave compared to other months. Clam mortality during the heatwave was likely a combined result of physiological death and increased predation. When put into the context of air temperature during fishing, there was a clear ecological shift at 30 °C, whereby clam reburrowing plummeted, and predator/scavenger activity and clam mortality dramatically increased. These results provide in situ documentation of human-climate interactions influencing indirect fishing mortality and altering ecological dynamics, ultimately generating pertinent information for ecosystem-based fisheries management.

Recent decades have seen increases in the frequency, severity, and duration of heatwave events globally—a trend which is projected to worsen as the climate continues to change[1]. These heatwaves can have substantial socio-economic and socioecological consequences, including extreme droughts and wildfires[1,2], heat-related mortality events in humans and other organisms[3,4], and severe socioeconomic impacts[5]. Such effects are not restricted to the terrestrial realm. Marine heatwaves—generally defined as discrete periods of anomalously high seawater temperatures—have also been increasing in frequency and severity[6,7]. Like terrestrial heatwaves, marine heatwaves have been linked to varied effects on marine organisms[8–11] and can have detrimental socioeconomic ramifications[12].

Terrestrial and marine heatwaves do not necessarily occur in concert, and their biological effects are generally limited to species residing within those respective habitats. For species living at the land-sea interface (i.e., littoral zone), however, both terrestrial and marine heatwaves can pose a threat. This is

perhaps most evident for intertidal species, where organisms are exposed to air during low tide and submerged in seawater during high tide. Many intertidal organisms reside at or near their physiological thermal tolerance limits[13]. Although air exposure is a stressor in and of itself[14], abnormally high air temperatures during low tide can compound thermal stress, while the return of cooler seawater at high tide can provide relief (though nuances are apparent depending on the movement and speed of both air and water, and the degree of shading)[15]. During heatwaves, particularly in coastal waters with small tidal ranges, extremely high air temperatures can impose concurrent high shallow water temperatures that may exceed species tolerance limits[16]. While behavioural thermoregulation can allow mobile organisms to deal with high temperature stress[17], sessile and semi-sessile organisms must endure such conditions without the ability to behaviourally avoid high temperatures. Consequently, contemporary heatwaves have led to mass mortalities of sessile nearshore organisms[4,18,19]. Even when mass mortalities do not occur, studies

[1]Fisheries and Oceans Canada, Moncton, NB, Canada. [2]Department of Biological Sciences, University of New Brunswick, Saint John, NB, Canada.
✉e-mail: jeffery.clements@dfo-mpo.gc.ca

suggest that heatwaves can result in broad ecological changes in nearshore systems[20–24].

While studies suggest that heatwaves can have ecological effects, adequate in situ field studies in nearshore systems are rare[25]. Such field studies are necessary to identify ecological mechanisms that may drive population- and community-level changes imposed by climate change, and to predict and adapt to these changes. Moreover, although some studies exist[26,27], research on the potential ecological effects of anthropogenic disturbances during heatwaves is limited. The act of fishing, for example, can result in ecological shifts by reducing prey abundances through the removal of targeted and non-targeted (bycatch) species[28]. While this knowledge has underpinned calls for ecosystem-based fisheries management since the 1990s[28], the ways in which heatwaves might influence the ecosystem effects of fishing remains understudied. As such, an understanding of how fishing during heatwaves influences ecological dynamics and fishing mortality is paramount for developing effective ecosystem-based fisheries management as the climate continues to change[28,29].

One globally-common nearshore fishery that has the potential to alter ecological function is clam digging. Clam digging during low tide can be destructive, as it directly removes infaunal organisms from their habitat, exposing them to potentially-lethal air temperatures. This is particularly apparent in clam fisheries regulated by legal size limits, where sub-legal sized clams are typically tossed back onto the sediment surface after being fished. For many species, reburrowing rates following such disturbance can be slow and environmentally dependent[30], thus increasing their vulnerability to predation and mortality[31,32]. Given that predator activity and consumption rates generally increase under higher temperatures[33,34], increased vulnerability to predation facilitated by human disturbance may be compounded during heatwaves, particularly if predator avoidance behaviours decrease in prey[24]. Consequently, anthropogenic disturbances such as clam digging have the potential to drastically alter interspecific interactions and ecosystem function during extreme heatwaves.

Eastern North America has experienced one of the strongest increases in heatwave frequency since the early 1980s[1]. This has been evident in temperate coastal regions such as Atlantic Canada, where a climate change-driven heatwave in June 2024 saw three consecutive days of air temperatures >10 °C above average (see "*Extreme weather event attribution*" from https://www.canada.ca/). In this study, we leverage an in situ field mesocosm experiment conducted during and outside of the June 2024 heatwave to address knowledge gaps related to the effects of anthropogenic disturbance during extreme heatwaves on ecological interactions at the land-sea interface. Our results show that clam digging during heatwaves can alter predator-prey and scavenger activity, and drastically increase indirect mortality of vulnerable sub-legal sized clams after they have been fished and tossed back to the sediment surface by clam diggers. Ultimately, this study reports strong field-based evidence for the potential of human-facilitated shifts in ecosystem function during heatwaves, and provides pertinent information for ecosystem-based approaches to fisheries management[35].

## Results

Air temperature data for the study site (Loggiecroft Beach, Kouchibouguac National Park, New Brunswick, Canada) are presented in Fig. 1. Air temperature data (maximum and mean air temperatures, and humidex temperatures) were compiled from historical data recorded by Environment and Climate Change Canada (https://climate.weather.gc.ca) at the Kouchibouguac weather station (≈8 km southwest of our study site). Generally, air temperatures increased from May through June, remained relatively stable for July-August, and started to decline near the end of August and into September (Fig. 1). With respect to the experimental trials, however, temperatures were abnormally high during the June trial, with a mean air temperature of 32.6 °C during fishing (Fig. 1). The May trial had the lowest mean air temperature (25.0 °C), while the other three trials had similar mean air temperatures (July = 28.8 °C, August = 27.4 °C, and September = 28.6 °C). Humidex values during the trials showed a similar trend, with slight differences for the July-September trials. Again, the June trial had the highest

average humidex temperature (39.8 °C, with some hourly measurements exceeding 40 °C) and May had the lowest (28.5 °C) (Fig. 1). There were larger differences in humidex temperatures between the other three trials (compared to air temperature), with the July, August, and September trials having humidex temperatures of 30.3 °C, 35.0 °C, and 33.8 °C, respectively (Fig. 1).

We used Bayesian generalized linear mixed models (BGLMM) to test for independent and interactive effects of experimental trial, predator treatment (crab inclusion *vs.* exclusion), tide level (intertidal, shallow subtidal, deeper subtidal), and time since being fished and tossed back (24 h, 48 h) on clam burrowing and mortality rates. With respect to burrowing, the global BGLMM model revealed a significant independent effect of trial on the proportion of reburrowed clams ($\chi^2_4 = 17.30$, $p = 0.0017$); no other significant effects were detected (Table S1). Herein, burrowing proportions during the June heatwave trial were significantly lower than all other trials, regardless of predator treatment, tide level, or time (Fig. 2 and Table S2). There were no differences observed between any of the other trials (Fig. 2 and Table S2).

With respect to clam mortality, the BGLMM model uncovered significant trial × time since fishing ($\chi^2_4 = 12.80$, $p = 0.0123$, Table S1) and predator treatment × time since fishing interactions ($\chi^2_1 = 11.53$, $p = 0.0007$, Table S1). There was also a marginally non-significant three-way interactive effect between trial, predator treatment, and time since fishing ($\chi^2_4 = 8.40$, $p = 0.0781$, Table S1). We thus explored pairwise comparisons across relevant levels of all three factors. Pairwise comparisons between months for each time and predator treatment revealed that while the May, July, August, and September trials had statistically similar mortality rates regardless of predator treatment or time since fishing, the June trial generally had significantly higher mortality rates than all other trials (Fig. 2 and Table S3). The only exception to this trend was for the predator inclusion treatment after 48 h, where the higher June mortality rates were marginally non-significant, owing to slightly increased mortality rates in July, August, and September for the PI-48 h treatment (Fig. 2 and Table S3). Pairwise contrasts between predator treatments for each trial and time suggested that, after 24 h, the June, August, and September trials had significantly higher mortality rates in the predator inclusion treatment as compared to the predator exclusion treatment (Fig. 2 and Table S4). After 48 h, however, mortality rates in the June trial were similar between predator treatments, while predator inclusion plots had higher mortality rates after 48 h in the July, August, and September trials (Fig. 2 and Table S4). Finally, pairwise contrasts between time periods after fishing showed that mortality rates for the June trial were significantly higher after 48 h compared to 24 h in the predator exclusion treatment (Fig. 2 and Table S5); there was also a marginally significant difference between time points for predator inclusion cages in the August trial, where mortality rates were slightly higher after 48 h (Fig. 2 and Table S5).

The number of clams found eaten in the experimental trials was generally low, with the exception of the June heatwave trial, where the number of clams eaten was nearly an order of magnitude higher than all other trials (Table 1). Considering all individual clam mortalities, 87.1% were attributable to clams being eaten by crabs and/or snails (Table 1). The predator exclusion treatment after 24 h in the June trial deviated from this, where only 44% of mortality instances were attributed to clams being eaten compared to 72% in the predator inclusion treatment (Table 1); after 48 h, however, 95% and 99% of dead clams were visually eaten in the predator exclusion and inclusion treatments, respectively (Table 1).

To estimate predator activity, we counted the total number of crabs (invasive European green crabs, *Carcinus maenas* and native rock crabs, *Cancer irroratus*), and the total number of buckets containing omnivorous mudsnails (*Ilyanassa obsoleata*), during each trial. Given the low counts of predators actively observed in each of the experimental plots, we did not analyze these data statistically. Instead, we simply summed the total number of crabs, and the total number of plots with mudsnails, during each trial to provide a general estimate of predator activity. Estimates of predator activity were drastically higher during the June heatwave trial compared to all other trials. Herein, the number of crabs observed in the experimental plots in the

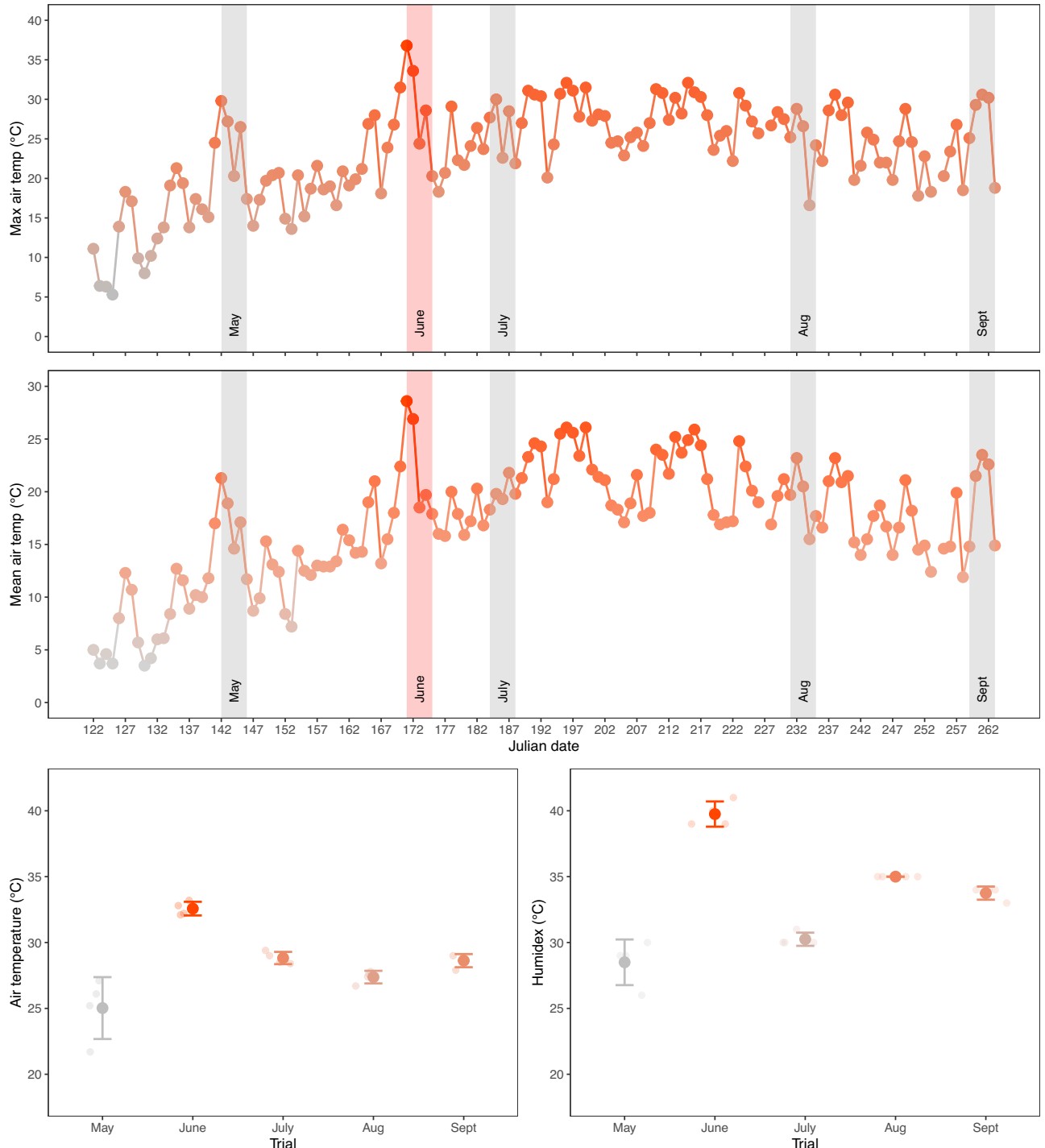

**Fig. 1 | Air temperature conditions during the experimental trials.** The top two panels show daily maximum (top panel) and average (middle panel) air temperatures from May–September 2024, with each point representing the value for a single day. Shaded areas in the top panels denote the timing and duration of each of the five, three-day mesocosm trials. See Supplementary Data 2 for source data for the top two panels. The bottom panels depict the mean ± standard deviation air temperature (left panel) and humidex temperature (right panel) during the

simulated 4 h fishing period on the first day of each trial. Data were obtained from Environment and Climate Change Canada's website for the Kouchibouguac weather station. Colours denote a gradient from lowest (gray) to highest (red) temperature. $n = 24$ measurements (1 per hour × 24 h) for each point in the top and middle panels; $n = 4$ measurements (1 per hour × 4 h) for each point in the bottom panels. Smaller points in the bottom panels are individual data points. See Supplementary Data 3 for source data for the bottom two panels.

June trial was nearly 2× higher than the experiment with the next highest crab count (Fig. 3). Likewise, the number of mesocosm plots containing mudsnails was nearly 4× higher in the June trial compared to the trial with the next highest count (Fig. 3). When counts of crabs and mesocosms containing mudsnails were summed to derive a "predator activity index", the June trial had an index >4× higher than the next highest trial (Fig. 3).

Finally, to place trial data into the context of air temperature thresholds, we plotted clam burrowing and mortality rates (proportions), and relative predator/scavenger activity (predator activity index, as above, relative to the highest observed index value for each trial), in relation to average air temperature during experimental fishing from each of the five trials. Loess curves were fit to the data to generate predictive relationships for each metric

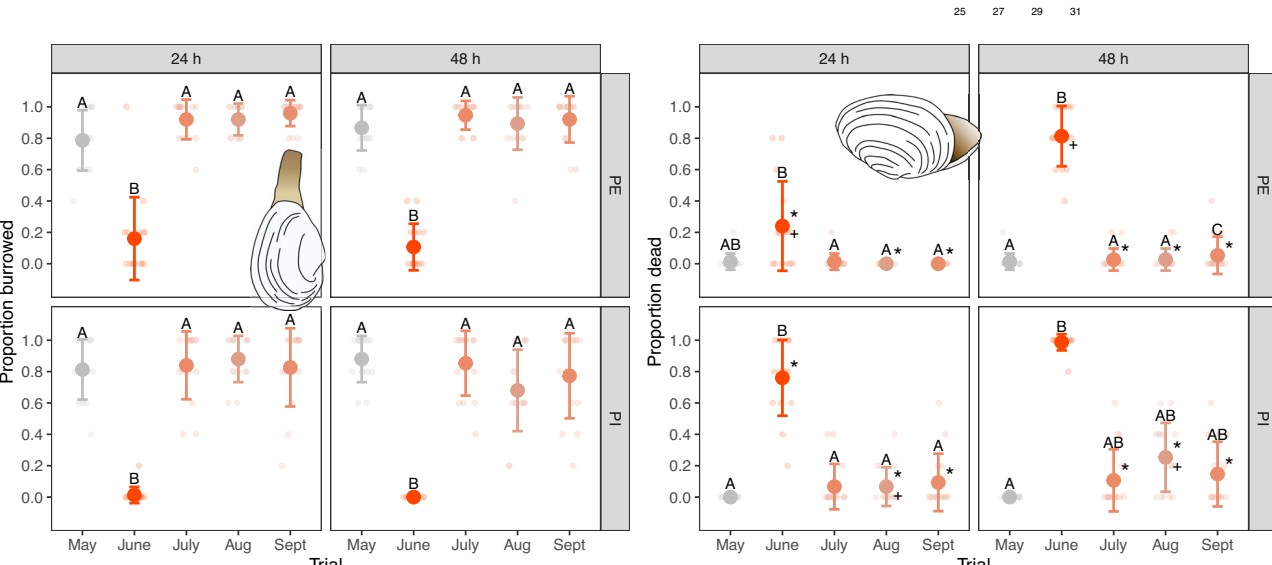

**Fig. 2 | Clam reburrowing and mortality rates.** Proportions of reburrowed (left panel) and dead (right panel) sub-legal sized clams after 24- and 48 h of being tossed back to the sediment after fishing in the predator (crab) exclusion (PE) and inclusion (PI) treatments for each of the five experimental trials. Capital letters above points denote pairwise differences between trials within each predator treatment and time point. Asterisks (*) beside points denote significant differences between predator treatments for a given trial and time point. Plus symbols (+) beside points denote significant differences between time points for a given trial and predator treatment. Data are means ± standard deviation. Colours denote a gradient from lowest (gray) to highest (red) mean air temperature during fishing. Smaller points are individual data points ($n = 15$ per trial × predator treatment × time combination). See Supplementary Data 1 for source data.

across an air temperature gradient. We also used generalized additive mixed modelling (GAMM) to directly test for effects of air temperature during fishing, predator treatment, tide level, and trial on clam reburrowing and mortality; models included Julian date as a random factor. The GAMM models revealed highly significant effects of air temperature on clam reburrowing and mortality (Table S6). There were no significant effects of any other fixed factors on reburrowing, which is consistent with the BGLMM models (although there was a significant effect of Julian date; Table S6). Similarly, the GAMM model results for the effects of other fixed factors on clam mortality were consistent with the BGLMM model results. Herein, there was a significant interactive effect of predator treatment × time since fishing, as well as significant independent effects of those two factors (Table S6). There was a clear ecological tipping point at air temperatures ≈30 °C, after which reburrowing rates plummeted and relative predator activity and mortality rates increased dramatically (Fig. 4). The relationships suggested that clam burrowing and mortality rates decrease and increase, respectively, by ≈50% between 31 and 32 °C; complete mortality and lack of burrowing are observed between 32 and 33 °C (Fig. 4). The relationship of increasing mortality rates with increasing air temperature was paralleled by increases in predator activity, opposite to the burrowing rate relationship (Fig. 4).

## Discussion

In this study, we document the interactive impacts of human disturbance (fishing) and climate change (heatwaves) on ecological interactions in a nearshore intertidal community in the context of an understudied commercial and recreational fishery. We found that fishing during extreme heatwaves can lead to increased thermal stress for sub-legal catch during fishing and after being tossed back. This increased stress led to reduced predator avoidance behaviour (reburrowing) and increased physiological mortality. At the same time, increased predator and scavenger activity was observed during the heatwave, and the number of clams found eaten in the experimental plots during the heatwave was exceptionally high compared to all other experimental trials. Ultimately, these results suggest that clam digging during heatwaves can increase indirect fishing mortality in tossed

back sub-legal clams by increasing thermal stress and physiological mortality, while simultaneously reducing predator avoidance behaviour and increasing predation/scavenging rates in surviving clams. The combined effects of thermal stress and increased predation/scavenging can ultimately drive near complete indirect fishing mortality during heatwaves (when indirect mortality is otherwise very low). These results provide strong, field-based evidence for human- and climate-facilitated ecological shifts during heatwaves, and provide key insights for ecosystem based approaches to fisheries management in a warming climate.

While laboratory studies of climate change effects on ecosystem function are relatively common, field studies directly documenting the ecological impacts of climate change are rare and often inadequate[25]. Furthermore, to our knowledge, direct in situ observations of heatwave-altered ecosystem function at the land-sea interface remain undocumented in the literature, particularly in the context of ecological shifts driven by human activity. Our study provides compelling field-based data to fill these knowledge gaps.

The results of this study provide convincing data pertaining to the roles of physiology and behaviour in mediating ecological function during extreme heatwaves. During the June heatwave trial, clams visually appeared to be in poor physiological condition when being tossed back, which likely led to the severely reduced burrowing rates observed during the heatwave. Although our data cannot isolate the specific physiological mechanisms leading to poor condition in the clams, there are a number of factors related to clam fishing practices that could negatively affect clam physiology. Our fishing methodology placed clams in buckets of seawater once they were fished. As such, during heatwaves, the water within the collection buckets would heat rapidly during fishing. This would drastically and quickly increase thermal stress on the clams, which would likely influence physiological condition. Given the observed air and humidex temperatures during the June trial (≈31–32 °C), it is likely that water temperatures in the buckets could have exceeded the lethal temperature for adult *M. arenaria* (≈30 °C)[36]. At the same time, clams kept in a five-gallon bucket during such high temperatures likely would have experienced increased respiration rates[37], potentially depleting oxygen in the buckets rapidly. While regular water changes were applied in our fishing method, this may not have been

## Table 1 | Mortality event data

| Experimental trial | Pred treat | Released | Time | Dead | Eaten |
|---|---|---|---|---|---|
| May (25.0 °C) | PE | 75 | 24 h | 1 (1.3%) | 0 (−) |
| | | | 48 h | 2 (1.3%) | 0 (−) |
| | PI | 75 | 24 h | 0 (0.0%) | 0 (−) |
| | | | 48 h | 0 (0.0%) | 0 (−) |
| June (32.6 °C) | PE | 75 | 24 h | 18 (24.0%) | 8 (44.4%) |
| | | | 48 h | 61 (81.3%) | 58 (95.1%) |
| | PI | 75 | 24 h | 57 (76.0%) | 41 (71.9%) |
| | | | 48 h | 74 (98.7%) | 73 (98.6%) |
| July (28.8 °C) | PE | 75 | 24 h | 1 (1.3%) | 0 (−) |
| | | | 48 h | 1 (1.3%) | 1 (−) |
| | PI | 75 | 24 h | 5 (6.7%) | 5 (100.0%) |
| | | | 48 h | 10 (13.3%) | 10 (100.0%) |
| Aug (27.4 °C) | PE | 75 | 24 h | 1 (1.3%) | 0 (−) |
| | | | 48 h | 6 (8.0%) | 5 (83.3%) |
| | PI | 75 | 24 h | 2 (2.7%) | 2 (−) |
| | | | 48 h | 19 (25.3%) | 19 (100.0%) |
| Sept (28.6 °C) | PE | 75 | 24 h | 0 (0.0%) | 0 (−) |
| | | | 48 h | 7 (9.3%) | 7 (100.0%) |
| | PI | 75 | 24 h | 0 (0.0%) | 0 (−) |
| | | | 48 h | 15 (20.0%) | 14 (93.3%) |
| Overall | | 750 | | 279 (37.2%) | 243 (87.1%) |

The number of released, dead and visibly eaten clams for each predator treatment and each time point during each of the five experimental trials. Temperatures under the 'experimental trial' column represent average air temperatures during fishing for each trial. Percentages under the 'dead' column represent the percentage of released clams that were dead. Percentages under the 'eaten' column represent the percentage of dead clams with tissues that were visibly consumed. For the predator treatment ('pred treat') column: PE predator (crab) exclusion, PI predator inclusion. 'Eaten' percentages are only provided for instances with ≥5 dead clams.

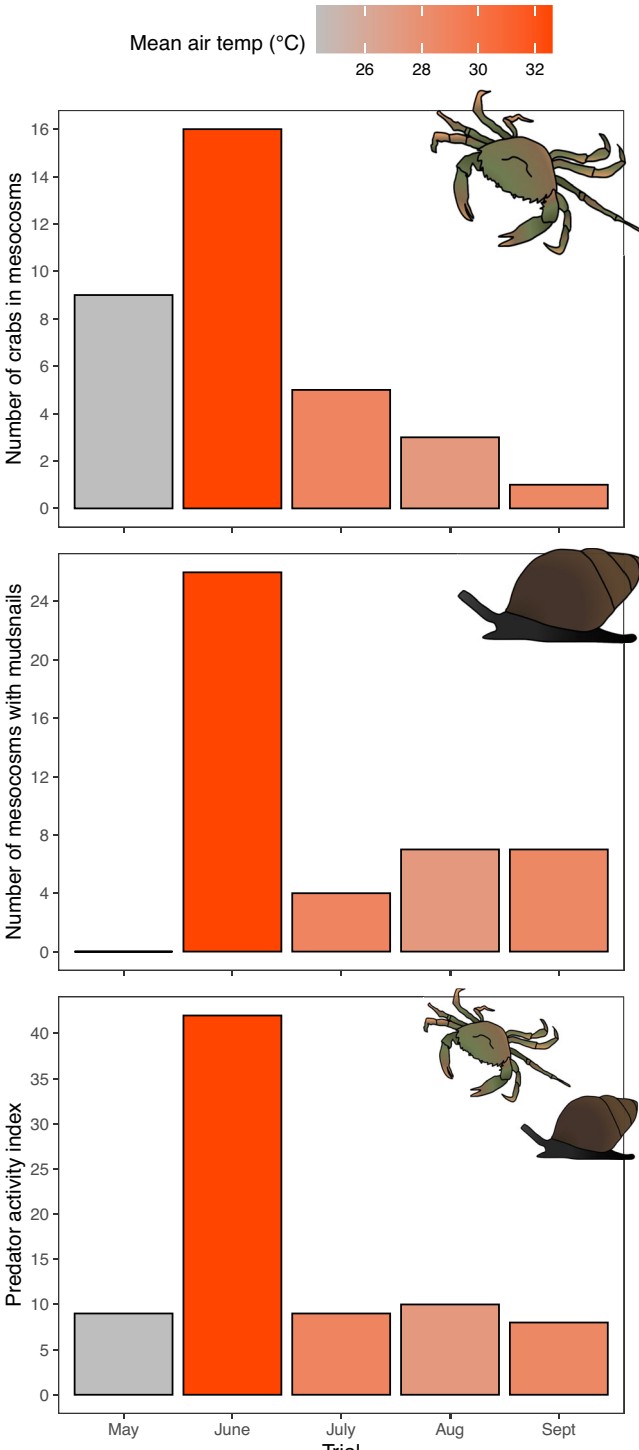

**Fig. 3 | Estimates of predator/scavenger activity during each of the five experimental trials.** The total number of individual crabs (*Carcinus meanas* and *Cancer irroratus*) (top panel), the total number of mesocosm plots containing omnivorous mudsnails (*Ilyanassa obsoleta*) (middle panel), and the overall predator activity index (crabs + mudsnail plots; bottom panel) observed across all mesocosm plots ($n = 30$) in each of the five trials. Colours denote a gradient from lowest (gray) to highest (red) mean air temperature during fishing. See Supplementary Data 4 for source data.

sufficient to avoid low oxygen conditions in the buckets, which may have further negatively affected physiology and behaviour. Indeed, exposure to sustained hypoxia under highly-controlled laboratory conditions has been reported to reduce burrowing behaviour in this species[38,39]. However, low oxygen cannot explain the observed increase in predator activity during the June trial, which is most likely a consequence of higher temperatures and/or the chemical attraction of predators by dead and dying clams. Unfortunately, we did not measure water temperature or oxygen in the buckets directly. Finally, in our experiment, clams were exposed to air for a short duration while washers and fishing line were glued to their shells. Air exposure during periods of high temperatures has been reported to increase physiological stress and mortality in other bivalves[40,41]. As such, exposure to high air temperatures during this time likely had negative effects on physiology and survival in the June trial. Ultimately, while we cannot pinpoint proximate causes of reduced burrowing and non-predation related mortality, it is very likely that high temperatures, low oxygen, and/or some combination of these stressors during fishing negatively affected clam physiology, leading to poor physiological health, increased vulnerability to predators, and contributing to the observed ecological shifts during the heatwave. Our results align with other recent findings that predator avoidance behaviour is reduced during heatwaves, which can alter ecological interactions between predators and their prey[24].

It is important to note that while our experiment attempted to mimic typical clam digging methods in the region, some aspects of our experiment deviated from what clam diggers would typically do. Notably, when digging for clams, clam diggers would not tether washers to their sub-legal catch as

**Fig. 4 | Predicted relationships with air temperature during fishing for clam reburrowing, clam mortality, and predator activity.** Curves generated from applying a Loess fit (step = 1) to raw burrowing and mortality proportions, at 48 h, for the average air temperature during fishing for each experimental trial (*n* = 30 per temperature; see Supplementary Data 5 for source data), and to raw individual values of relative predator activity (i.e., predator activity index as in Fig. 3, relative to the maximum observed index value; *n* = 1 per temperature; see Supplementary Data 4 for source data). Shaded areas are 95% confidence bands.

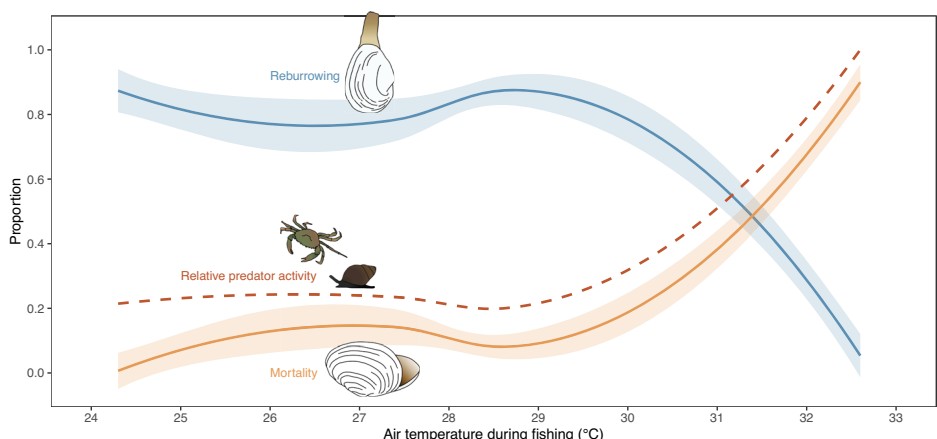

we did in our experiment. Unfortunately, clam digging method is something that is difficult to control for because not all clam diggers use the same methods when fishing for clams, as there are no standardized management rules for how to handle/hold clams while fishing. For example, some clam diggers will put the clams in a bucket of water, while other diggers do not add water, but just keep the clams in a dry bucket because filling it with water makes it heavy and would increase physical fatigue; if using water, some clam diggers would change out this water regularly as we did while others would not. Additionally, some clam diggers will measure and toss clams back on-the-fly while others will wait until they are finished fishing. While our methods do not replicate all of the ways in which clam diggers fish for clams, clam diggers most often fish for the maximum possible duration that they can during low tide (typically 5–6 h; well beyond our fishing times), particularly in the southern Gulf of St. Lawrence because tide cycles are semi-diurnal with small tidal ranges (days where clam diggers can fish are limited throughout the year). Anecdotally, during one of the trials, we observed a recreational clam digger fish for clams for a longer duration than our fishing and tethering process, place all clams in a bucket without water, and measure and toss back the sub-legals once they were finished digging. Such an approach during a heatwave would expose the fished clams to both desiccation and extreme air temperatures, likely maximizing physiological stress. As such, it is unlikely that our fishing methodology would invoke more stress on the clams than would be invoked by the practices of many clam diggers. Nonetheless, there may exist some optimal clam digging method that could reduce stress on the animals when fishing during heatwaves or other acute environmental stresses. Further studies exploring the influence of different holding conditions when clam digging during periods of environmental stress on sub-legal burrowing and survival are thus warranted.

Although effective at minimizing crab predation, our exclusion plots did not keep 100% of crabs out[30]. As such, while it is clear from our results that fished sub-legal clam mortality increases during heatwaves, we are unable to determine the relative contributions of physiological death, predation, and scavenging in causing mortality with a high degree of precision. Nonetheless, we can make some inferences based on observed mortality rates and the proportions of eaten vs. uneaten clams in the respective treatments (Table 1) to generate approximations. In the predator exclusion treatments, clam mortality after 24 h in the June trial was 24%, with 44% of those clams (≈10% of all clams) found with tissues eaten. Unfortunately our predator exclusion cages did not keep 100% of predators out. However, conservatively assuming that all clams in the predator exclusion cages died physiologically, and that eaten clams were scavenged upon, suggests that, in the absence of predation, 14% of clams physiologically die and are not scavenged, 10% physiologically die and are scavenged upon, and 76% survive. In contrast, 24 h June mortality in the predator inclusion treatments was 76%, with 72% of those dead clams (55% of total clams) being found with their tissues eaten. Interestingly, the 28% of dead clams (21% of all clams) that

were uneaten in the predator inclusion treatments closely approximates the inferred 24% physiological mortality rate (with and without scavenging) in the predator exclusion treatments as above. This suggests that, when predators are present, scavenging is reduced at 24 h and the majority of eaten clams are likely consumed by predators; if scavenging was occurring, we would expect the portion of uneaten dead clams in the predator inclusion treatment to closer approximate the 14% of clams that physiologically died but were not scavenged in the predator exclusion treatment. As such, we can estimate that, when predators are present, ≈55% of tossed back sub-legal clams are likely consumed by predators, ≈20–25% succumb to physiological death, and ≈20–25% survive 24 h after fishing during extreme heatwaves; however, when predators are largely absent, ≈20–25% of clams succumb to physiological death after 24 h, with ≈45% of those (≈10% of all clams) scavenged upon. Interestingly, at 48 h after fishing, the overall mortality rate of clams in the predator exclusion treatments increased tremendously, from 24% (at 24 h) to 81% (+53%), with 95% of those dead clams being eaten. Since predator exclusion treatments excluded most (but not all) predators, this suggests a high rate of physiological death and scavenging 48 h after being tossed back to the sediment. While crabs can certainly scavenge on dead clams, we also found increased activity of mudsnails (*Ilyanassa obsoleta*) in the mesocosm plots during the June heatwave trial, which are known to both scavenge on dead molluscs[42,43] and actively predate on dying and unhealthy molluscs[44]. Indeed, we directly observed congregations of mudsnails consuming tissues of clams in our June trial. Similarly, mortality in the predator inclusion treatment increased from 76% at 24 h to 99% at 48 h (+23%). Given the drastic increase in mortality in the predator exclusion treatment at 48 h, coupled with the presence of both live and uneaten clams in predator inclusion treatments at 24 h, it is probable that this increase in mortality rate between 24 and 48 h in the predator inclusion treatments was also due to physiological death and scavenging, rather than predation. This ultimately indicates that while predation can serve as a proximate cause of mortality in tossed back sub-legal clams within 24 h, the role of predation appears ephemeral and, even in the absence of direct predation, the majority of clams are likely to succumb to physiological death and be scavenged after 48 h. Nonetheless, our results provide strong evidence for ecological shifts driven by human-climate interactions that are likely to have ramifications for the flow of energy throughout the ecosystem, as observed in other systems[7,45]. Relative abundances of different predators, scavengers, and prey within a given system will influence heatwave effects on energy flow.

The findings herein align with previous studies suggesting that, generally, predator activity and consumption increases under higher temperatures[33,34]. Recently, it has been broadly suggested that consumption rates of benthic predators increase during marine heatwaves[7]. Our results provide direct empirical support for this claim, as both predator activity and the number of clams found eaten increased during the June heatwave trial. Furthermore, data from the crab monitoring program also point to

heatwaves increasing predator activity, as crab catches during June 2024 (heatwave) were ≈2× higher than those of the same time period for the five preceding years (2019–2023) (Fig. S1). What is unknown from our experiment, however, is whether being burrowed confers an adequate predator avoidance strategy for clams, as the mortality rates reported in our experiment were facilitated by human activity (clam digging). Nonetheless, given the near complete mortality of sub-legal sized clams observed during the heatwave and the fact that almost all of the clams were found eaten, it is clear that the combined effects of both human activity and climate change can intersect to drastically alter ecosystem function.

One potential pitfall of our study is that our experiment was conducted at a single site and replicated biological observations through time over five separate experimental trials. As such, our design lacks a comparative control treatment with experimental trials conducted at the same time in the absence of a heatwave. Indeed, this is perhaps one of the reasons why in situ climate change experiments are rare, as deploying a suitable control treatment during climate events such as heatwaves is highly challenging. As such, it is possible that the observed changes in the June experiment were driven by factors other than the heatwave. For example, predictable seasonal shifts in predator-prey activity as Spring transitions to Summer may have driven the reduced burrowing and increased predator activity in June. With respect to predator activity, data spanning 2019–2024 from a crab monitoring program revealed no predictable increase in predator activity between late May and early June as observed in our experiment (Fig. S1). Interestingly, 2024 data from the crab monitoring program followed a similar trend in crab catches as the estimates of predator activity recorded in our experimental plots, with crab catches for the month of June being 2x higher in 2024 than any of the five preceding years. Together, our data and that of the crab monitoring program strongly suggest that the observed increase in predator activity in the June trial of our experiment were indeed driven by the heatwave. Similarly, comparing the reburrowing rates of sub-legal clams at the same site between two years revealed that reburrowing rates do not predictably plummet during similar dates in June, but rather appear to be influenced by acute environmental changes such as heatwaves and low salinity events (Fig. S2). However, we did not observe any evidence that an acute salinity change could have driven the observations in our June trial, as there was no rainfall observed nine days before, nor during, the June experimental trial (Fig. S3), and salinity effects only appear after extreme rainfall events (e.g., 70+ mm in <24 h[30]). Coupled with these additional data sources, our results strongly suggest a causal relationship between fishing clams during the extreme heatwave in June and the observed shifts in predator-prey activity, and increased sub-legal clam mortality.

Ecosystem-based approaches to resource management constitute adaptive management strategies that seek to consider the entire ecosystem, including humans, in resource-based decision making[46,47]. This management approach not only considers the conservation and sustainability of the targeted resource, but attempts to ensure the sustainability and conservation of the ecosystem as a whole[48,49]. While the idea of ecosystem-based management is not new[50], global fisheries have long managed stocks by considering the target species in isolation, and the uptake of ecosystem-based management has been slow[51]. Furthermore, while it is important to integrate climate change impacts, such as heatwaves, into ecosystem-based management approaches[52], studies doing so are rare (although some do exist)[53,54]. Our results represent a key example of why it is critical to implement ecosystem-based approaches to fisheries management, particularly in a time of rapid climate change. When clam burrowing and mortality rates, and predator activity rates, from our experiment were put into the context of air temperatures during fishing, we observed a clear ecological tipping point ≈30 °C air temperature, after which reburrowing rates of sub-legal, tossed back clams plummeted, predator activity increased, and the sub-legal clams suffered near complete mortality. As such, the management of nearshore clam fisheries globally—and any fishery occurring at the land-sea interface —should consider the consequences of fishing during periods of high temperatures in their management frameworks. Some regional fisheries already do this. For example, some catch-and-release Atlantic salmon

fisheries in eastern Canada are partly managed by a "warm water protocol", whereby angling is restricted at water temperatures exceeding particular thresholds that can lead to high angling-related mortalities[55]. Following such logic, our results suggest that resource managers responsible for clam fisheries in Atlantic Canada may want to consider conservative management options when air temperatures exceed 30 °C (Fig. 4). Furthermore, our results underscore the need to consider heatwaves and other periods of abnormally-high temperatures for a wide range of coastal fisheries. How do heatwaves impact the probability of survival in other catch-and-release fisheries, or the survival of bycatch in other fisheries? Given that heatwaves are posing an increasingly-frequent threat on Earth's ecosystems, the application of environmental parameters and ecosystem interactions (including humans) into resource management decisions is imperative for the future of biological conservation.

## Methods

### Study site
Experimental trials were conducted in the intertidal zone at Loggiecroft Beach, located in Kouchibouguac National Park, New Brunswick, Canada (46.835087°N, 64.932245°W). The park's marine habitat is part of the Gulf of St. Lawrence and consists of two large lagoons protected by a series of barrier islands. There are a total of 21 recreationally-fished soft-shell clam beds within the park that are monitored for clam densities semi-annually[56]. Our study site was selected based on ease of access, good habitat quality, ideal sandy-mud sediment conditions[57], and high contemporary clam densities providing for efficient and productive fishing[30,56,58].

### Experimental protocol and design
A total of five experimental trials were conducted over the course of five months (May to September 2024), spanning the recreational soft-shell clam (*Mya arenaria*) fishing season at Kouchibouguac National Park (see "*Clam fishing*" from https://parks.canada.ca/pn-np/nb/kouchibouguac). Exact dates of the experimental trials were: May 22–24, June 20–22, July 3–5, August 19–21, and September 16–18. Soft-shell clams were chosen as the focal clam species because they are a common intertidal species distributed circumglobally across northern temperate latitudes. This species constitutes sizable recreational and commercial fisheries in eastern North America (Canada and the USA)[59], including the region where this study took place (2023 southern Gulf of St. Lawrence commercial soft-shell clam landings = 378,965 metric tons valued at CA$1,557,383[60]). The timing of experimental trials was scheduled to coincide with optimal low tide times (i.e., <0.3 m, measured against lower low water large tide) predicted for the nearest tide station available from Fisheries and Oceans Canada (see "*Pointe Sapin – 01830*" from https://tides.gc.ca/en/stations/). Each trial was conducted over a period of three days: one day for setup, fishing clams, and initial burrowing observations, and the other two days for data collection.

The first day of each experimental trial was dedicated to setting up experimental plots and fishing, tethering, and releasing sub-legal sized soft-shell clams. A team of three-four people was deployed during this day. Upon arrival at the study site, two-three team members began fishing clams as soon as the tide was low enough by using drain spades to dig out patches of sediment, targeting areas with many visible siphon holes[57]. One other team member would measure the shell length of the fished clams using digital calipers. Clams >50 mm (minimum legal size limit for recreationally-fished soft-shell clams in the southern Gulf of St. Lawrence, including Kouchibouguac National Park; see "*Conservation Measures for clams in eastern New Brunswick, Gulf Nova Scotia and Prince Edward Island*" from https://www.glf.dfo-mpo.gc.ca/) or <30 mm shell length were returned to the subtidal sediment surface to maximize survival[30], while sub-legal sized clams (30–50 mm shell length; see Fig. S4 for size distributions in each trial) were retained in a five gallon bucket filled with seawater after their shell length was recorded. This size range was chosen because previous work showed that burrowing rates of clams at our study site are unaffected by shell length across this size range[30]. Holding conditions aimed to mimic methods used by many clam diggers (based on personal conversations with a few clam diggers prior to

experimentation; although not all clam diggers fill their buckets with water). Holding conditions were also designed to minimize physiological stress associated with air exposure while the clams are out of their burrows. Clams were fished and measured until a total of 150 sub-legal clams were caught. Precise fishing times varied between trials, typically ranging between 2.5–3.5 h in duration; seawater in the buckets containing retained clams was regularly changed every ≈45–60 min to minimize temperature and oxygen stress. Once 150 sub-legal clams were obtained, water was changed in the bucket and three team members removed the clams from the bucket of seawater, dried them with a paper towel, and attached a visible tether (≈20 cm of fishing line attached to a flat metal washer; ≈0.35 g) to each individual clam's shell using Solarez® UV resin (as per the methods described in Ledoux et al.[30]), inside a vented UV-blocking tent. Metal washers were chosen over plastic washers because preliminary experiments showed that plastic washers floated and lifted the clams upright off the sediment surface, inhibiting them from burrowing. While metal washers are heavier, the degree of slack provided by the fishing line ensured that clam burrowing was not impeded by the weight of the washer. Once the tether was glued, the clam was placed on the sand for UV resin to dry while the remainder of the clams were tethered (30–60 min). Once tethers were attached to all 150 clams, the clams were randomly placed in experimental mesocosm plots ($n = 5$ clams plot$^{-1}$). The total time from the start of fishing to the start of the experiment (i.e., clams placed in experimental plots) ranged from a minimum of ≈3.5 h in the May trial to a maximum of ≈4.5 h in the July and September trials (Table S7). These fishing times are well within the daily time frames that clam diggers would be harvesting (typically 5–6 h per day).

Experimental mesocosm plots were constructed from five gallon buckets (The Home Depot™). Two different types of plots were deployed in our experiment: predator exclusion (PE) plots, and predator inclusion (PI) plots. Predator exclusion plots were constructed as per the methods of Ledoux et al.[30]. Briefly, the bottom half of each bucket was removed using a jigsaw to allow the bucket to be pushed into the sediment, and a series of small holes (<1 in diameter) were drilled in the sides of the buckets to allow water to move freely in and out of the buckets without increasing water pressure and disturbing clams on the surface. The lids of the predator exclusion plots were hollowed out by cutting out the middle surface with a jigsaw such that only the rim of the lid remained. After clams were placed in the predator exclusion plots, the top of the bucket was covered with a 0.5 mm mesh and the lid was secured into place over the mesh. While the predator exclusion plots do not exclude all predators (i.e., milky ribbon worms, moonsnails, small epibenthic predators) or scavengers (mudsnails, shrimp), our previous work showed that predation-related mortality in these plots was low (<1[30]). Furthermore, of all crabs observed in plots during the experiment, 75% were in PI plots, with those in PE plots restricted to very small sizes. We also did not see any live moonsnails or milky ribbon worms in any of our trials, nor any of the previous work we've conducted at this site. Predator inclusion plots were also constructed from five gallon buckets. The PI plots were cut shorter than predator exclusion plots such that there was only ≈3 inches of bucket above the sediment surface, and mesh lids were not used to cover the top of the bucket, thus allowing epibenthic predators access to the clams inside.

After tethering was complete, the mesocosm plots were deployed adjacent and upstream to where clams were fished (≈5 min). Plots were placed in a spatially randomized order at three pre-determined tide levels: intertidal, shallow subtidal (just beyond the low tide water line), and deeper subtidal (≈10 feet beyond the shallow subtidal level) (Fig. S5). A total of five plots from each predator treatment were randomly assigned at each tide level ($n = 10$ plots per tide level, 30 plots per experimental trial). All plots were secured in the field by pushing them into the sediment until they were generally stable and then affixing them, using cable ties, to two lengths of rebar hammered into the sediment. Once plots were in place, five tethered clams were placed in each plot; a sample size of five clams per plot was chosen to avoid overcrowding the mesocosm plots and introducing clam density effects on reburrowing behaviour. The number of clams that reburrowed was recorded every 15 min for 2 h (120 min) thereafter to validate the results of Ledoux et al.[30] ($n = 8$ time points on Day 1); no clam

mortality was observed on the first day of any of the five experimental trials. A reburrowed clam was defined as a clam that was observed in a naturally-burrowed position (i.e., siphons upright) with >90% of its length beneath the sediment surface. A dead clam was defined as a gaping clam with its siphon partly or fully extended that did not respond (i.e., move or close shell) to three sequential tactile stimuli (i.e., poking with a finger), or as a tethered clam with its tissue either partly or fully consumed by a predator or scavenger. Observations of burrowing and mortality could not be blinded due to distinct visual differences between predator inclusion and exclusion cages and the obvious nature of each tide level. However, all observations of clam burrowing were made by at least two field team members to avoid individual researcher bias (confirmation bias). If there were discrepancies between team members regarding whether a clam was burrowed, an additional team member would provide their assessment and the team would collectively determine if a clam was or was not considered burrowed. Given that no mortality was observed on Day 1 and we were interested in burrowing and survival in the further term (i.e., 24 and 48 h after clams were tossed back), we do not depict Day 1 reburrowing results in the main paper; however, statistical methods, results, and interpretation are provided in the "Supplementary Note" file available on the Open Data Portal page (see Data Availability Statement for URL). Two team members returned to the field 24 and 48 h later and recorded the number of reburrowed and dead clams in each of the mesocosm plots; they also counted the number of crabs (green crabs, *Carcinus maenas*, and rock crabs, *Cancer irroratus*) that were observed in each mesocosm plot on each of the two days, as well as whether the plots contained scavenging mudsnails (*Ilyanasssa obsoleta*). After reburrowing and mortality observations were completed at 48 h, tethers were removed, experimental clams were returned to subtidal sediment, and experimental plots were removed, washed and stored inside until the next trial.

## Statistics and reproducibility

All statistical analyses were conducted in RStudio[61] using R version 4.4.0[62] with statistical significance set at $p \leq 0.05$. All plots were generated using the 'ggplot2' package[63]. Prior to analyses, proportions of reburrowed and dead clams were computed from raw counts by dividing the number of reburrowed and dead clams by the number of clams initially deployed in each plot ($n = 5$). To facilitate reproducibility and transparency, which is particularly needed in the field of experimental biology[64], all supporting information for this article can be openly accessed in the Electronic Supplementary Material alongside this article, as well as the Government of Canada's Open Data Portal (see URL in Data Availability Statement). This supplementary material includes: 1. Supplementary Information (Figs. S1–S5, Tables S1–S7, and a Data Dictionary); 2. Supplementary Note (methods, results, and interpretation of Day 1 reburrowing as mentioned above); 3. Supplementary Data (Supplementary Datas 1–10; source data files used in analyses and figure generation); and 4. Annotated R Code.

Bayesian generalized linear mixed modelling (BGLMM) was employed to test for the independent and interactive effects of experimental trial (fixed categorical factor with 5 levels: May, June, July, August, September), predator treatment (fixed categorical factor with 2 levels: predator inclusion, predator exclusion), tide level (fixed categorical factor with 3 levels: intertidal, shallow subtidal, deeper subtidal), and time since being fished and tossed back (fixed categorical factor with 2 levels: 24 h, 48 h) on the proportion of dead and reburrowed clams; plot ID (unique within each trial) was included as a categorical random variable to account for random spatial effects and repeated measures across the two time points. A BGLMM approach was selected due to the near-complete statistical separation observed in our dataset (i.e., levels of predictor variables are completely separated by the response variable, which, in our case, occurs when certain levels of the predictor variables include all 0 counts of reburrowed or dead clams[65]). To account for this, one can use a Bayesian approach to set weak zero-mean normal priors on the fixed effects to account for such separation as suggested by Bolker (see "*supplementary materials for Bolker (2015)*" from https://bbolker.github.io/mixedmodels-misc/). As such, models were constructed using the *bglmer()* function from the 'blme' package[66], specifying a

binomial family (i.e., logistic regression for proportional data), zero-mean nominal priors (10, 60) on the fixed effects. A variance prior of 10 was selected due to the high degree of variance in the data driven by near-complete separation, and a diagonal matrix value of 60 was chosen to match the number of terms in the models. We first tried fitting the models with a variance prior of 9 (typical for datasets with near-complete separation); however, the PIRLS step-halvings for the mortality model failed to adequately reduce the variance in pwrssUpdate, suggesting that our variance prior did not capture the actual variance in the data. We therefore selected a slightly larger variance prior of 10, which alleviated the error, and applied the same variance prior to the burrowing model for model consistency. Notably, applying a variance of 9 or 10 for the burrowing model did not meaningfully alter model output, suggesting that results for the burrowing model are reasonable. The *Anova()* function from the 'car' package[67] was then used to obtain model results, specifying a Type 3 test for random effects models. Where significant main effects were detected, significant pairwise differences were determined using the *pairs()* function from pairwise models created using the *emmeans()* function with the 'emmeans' package[68], specifying a Holm adjustment.

It is important to note that evidence for a heatwave effect from the BGLMMs above is based on categorical time periods for experimental trials conducted at the same site. As such, evidence for heatwave effects on clam burrowing and mortality are restricted to temporal correlations between observed biotic responses and temperature conditions during those categorical timepoints. However, other environmental influences not related to temperature that vary over time could potentially influence any observed biotic responses. For example, low salinity events are known to drastically reduce soft-shell clam burrowing rates[30]. Similarly, seasonal shifts in predator and prey activity could also influence the observed results. To account for this, we supplemented the BGLMM approach above with additional analyses to more directly test for the effects of air temperature during fishing on clam reburrowing and mortality rates. Specifically, we built negative binomial generalized additive mixed models (one for reburrowing and one for mortality) using the *gamm()* function from the 'mgcv' package[69]. Models included average air temperature during fishing as a continuous smooth term, the independent and interactive effects of predator treatment, time since fishing, and tide level as categorical parametric terms, and Julian date as a continuous smooth random term.

### Exploring alternative explanations for differences between trials

To investigate potential seasonal effects on the observed trends in predator activity across the experimental trials, we leveraged weekly data from a crab monitoring program conducted near our study site by Kouchibouguac National Park from 2019–2024 (independent from the data in our experiment). This annual program, designed to monitor invasive European green crabs (*Carcinus maenas*) in the park[70], deploys a series of modified eel traps at various sites throughout Kouchibouguac National Park. From June-October, park staff fish the traps every 1–2 weeks, recording counts of all crab species found in the traps, as well as the number of days the traps were deployed. One particular site (named 'Loggiecroft') located near our experimental study site has been monitored consistently across years, and we were able to obtain data on crab catches and fishing effort back to 2019. We thus plotted average crab catch per day (total number of crabs ÷ number of days traps were deployed; average of $n = 2$–4 traps) recorded on each of the fishing days to observe temporal trends in crab catches from this monitoring program over the past six years (Fig. S1). We used these trends to descriptively compare crab monitoring trends to the estimates of crab activity in our experimental trials. If crab counts in our experimental plots were seasonally driven, we would expect to see similar temporal trends in crab catches in the monitoring data as we did in our experimental plots (i.e., increasing crab catches from early to late June with a sharp decrease and stabilization thereafter) across years.

To further investigate if the observed differences in burrowing and mortality between experimental trials could be attributed to seasonal cycles in clam burrowing rather than the June heatwave, we leveraged existing unpublished data from our lab group that is complementary to, but not included in, the results reported in Ledoux et al.[30]. In that study, we measured sub-legal soft-shell clam reburrowing times in predator exclusion cages at four different sites in Kouchibouguac National Park (including the same study site in this paper) during the summer of 2021, where a heatwave did not occur. Incidentally, this provided us with additional data on sub-legal clam reburrowing rates 24 h after being fished from June 21, 2021 (no heatwave) and four time points thereafter to which we could compare our 2024 experiment data to. While differences in experimental design preclude us from comparing the burrowing frequencies statistically (the 2021 study had multiple sites and only deployed three clams per plot), we nonetheless used the data to visually assess temporal trends in sub-legal clam reburrowing across two different years (Fig. S2). Herein, if trends in burrowing activity were seasonally driven, we would expect to observe a similar reduction in the percentage of clams reburrowed in the June trials for both years.

Acute changes in other environmental parameters, such as decreases in salinity following heavy rainfall events, have the potential to reduce sub-legal soft-shell clam reburrowing rates[30] and could thus drive observed differences between experimental trials. While we did deploy a salinity logger (Star-Oddi) at the study site over the duration of the experimental trials, the logger unfortunately malfunctioned and did not record any salinity data. In lieu of in situ salinity conditions, we retrieved daily precipitation data from the Kouchibouguac weather station (same source as the air temperature data) for the duration of the experiments to determine if any heavy rainfall events prior to the trial could have resulted in low salinity conditions during the trial. We then plotted daily precipitation totals for each of the five months of the experimental trials (May-September; Fig. S3). If the anomalous burrowing/mortality and predator activity estimates during the June trial in our experiment were driven by low salinity during the trial, we would expect to observe heavy rainfall event(s) in the day(s) leading up to the June trial, as well as a stark absence of heavy rainfall events leading up to the other experimental trials (where proportions of reburrowing were high and mortality low).

### Reporting summary

Further information on research design is available in the Nature Portfolio Reporting Summary linked to this article.

### Data availability

All source data for analyses and figure generation are openly available in the Electronic Supplementary Material alongside the online version of this article, as well as through the Government of Canada's Open Data Portal at https://open.canada.ca/data/dataset/1bf057da-8280-11ef-8cce-55cc7f028297. Supplementary materials include: 1. Supplementary Information (Figs. S1–S5, Tables S1–S7, and Data Dictionary); 2. Supplementary Note (methods, results, and interpretation of Day 1 reburrowing as mentioned above); 3. Supplementary Data (Supplementary Datas 1–10; source data files used in analyses and figure generation); and 4. Annotated R Code.

### Code availability

Annotated R code for analyses and figure generation are openly available in the Electronic Supplementary Material alongside the online version of this article as well as through the Government of Canada's Open Data Portal at https://open.canada.ca/data/dataset/1bf057da-8280-11ef-8cce-55cc7f028297.

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

## Acknowledgements

We wish to thank Dr. Daniel Gallant and Rémi Donelle of Kouchibouguac National Park for providing crab monitoring data, and Courtney Jones (Mount Allison University) for guidance and assistance with field work. Thanks to Dr. Luc Comeau (DFO Gulf), Dr. Jeff Barrell (DFO Gulf), and Prof. Fredrik Jutfelt (Norwegian University of Science and Technology and Gothenburg University) for constructive feedback on earlier drafts of this manuscript. We also want to sincerely thank two anonymous reviewers and the editor, whose extensive and constructive feedback drastically improved the manuscript. This project was funded by Fisheries and Oceans Canada through a grant from the Fisheries Science Collaborative Program awarded to JCC (project number 4T-039).

## Author contributions

J.C.C.: conceptualization, data curation, formal analysis, funding acquisition, investigation, methodology, project administration, resources, software, supervision, validation, visualization, writing–original draft, writing–review & editing. S.H.: investigation, data curation, funding acquisition, methodology, writing–review & editing. M.R.: investigation, methodology, funding acquisition, writing–review & editing. J.H.: investigation, methodology, writing–review & editing. B.L.P.: investigation, methodology, writing–review & editing. R.S.: conceptualization, funding acquisition, writing–review & editing.

## Competing interests

The authors declare no competing interests.

## Ethical approval

Sampling did not require ethics approval from the Canadian Council of Animal Care (CCAC), as the study species is not considered under their purview. Sampling and experimentation was conducted in accordance with federal regulations under DFO Section 52 License number SG-RHQ-24-076 (request no. 8) and Parks Canada Agency Research and Collection Permit number KOUCHNP-2024-458201.
