## [Transparent Peer Review file · Communications Biology]

Fishing during extreme heatwaves alters ecological interactions and increases indirect fishing mortality in a ubiquitous nearshore system

Corresponding Author: Dr Jeff Clements

This manuscript has been previously submitted to another journal. This document only contains information relating to versions considered at Communications Biology.

Version 0:

Reviewer comments:

Reviewer #1

(Remarks to the Author)

In the manuscript “Human activity during an extreme heatwave alters predator-prey dynamics in a ubiquitous nearshore system”, the authors conduct a field experiment to investigate how clam fishing activity can interact with marine heatwaves to reduce the burrowing behavior of released sublegal sized clams and increase their mortality rates. The effects of marine heatwaves have been an increasingly hot topic these past few years, and the findings from this experiment are dramatic and pertinent to a wide audience. Field experiments on the interactive effects of human activity and extreme events are relatively rare and difficult to conduct rigorously. The biggest issue with this study is that there is not a good control for the heatwave either by experimental manipulation of surroundings or use of alternative sites simultaneously. There is only one heatwave examined which overlaps with some seasonal timing and all before and after measurements occur at only one site which opens up the possibility of confounding factors. Given the magnitude of differences between heatwave and non-heatwave observations, most of the observed variation is likely to be attributed to the heatwave. However, it would be good for the authors to spend more effort refuting alternative hypotheses and more directly testing for the effects of the heatwave. Other issues are that the work needs to be better placed in the context of the broader literature as well as more thoroughly discuss/describe how their methods relate to and depart from standard clam fishing methods. Overall, this paper is well-written, interesting, and will be a good contribution to the literature.

Abstract

Lines 18-19: The wording on the second half of this sentence needs to be toned down. “Absent” is an exaggeration. For instance, Freedman et al. (2020) found that marine heatwaves had similar effects for kelp forest fish communities regardless of regional fishing activity while Zhang et al.’s (2022) data indicate heatwaves intensify the effects of nutrient loading and herbicides.

Freedman, R. M., Brown, J. A., Caldow, C., & Caselle, J. E. (2020). Marine protected areas do not prevent marine heatwave-induced fish community structure changes in a temperate transition zone. *Scientific Reports*, 10(1), 21081.

Zhang, P., Wang, T., Zhang, H., Wang, H., Hilt, S., Shi, P., ... & Xu, J. (2022). Heat waves rather than continuous warming exacerbate impacts of nutrient loading and herbicides on aquatic ecosystems. *Environment International*, 168, 107478.

Line 22-25: This sentence needs to be split in two.

Introduction

Lines 60-61: The authors need to be careful not to exaggerate. There is a lot of work showing how nearshore coastal systems respond to marine heatwaves with and without mass mortality (e.g. Spiecker and Menge 2022, Whalen et al. 2023).

Theoretical work on this subject is also common and lessons can be derived from other disasters that produce similar effects as marine heatwaves to estimate how ecological interactions can change. There are even a few studies that explicitly look at how ecological interactions shift in sessile nearshore systems due to marine heatwaves (e.g. Correia-Martins et al. 2022). The available literature widens when non-sessile organisms are included (e.g. Leung et al. 2019). The authors should instead focus on the fact that such field studies are rare and that they are necessary to identify the underlying mechanisms for how populations and communities will shift (to better predict and remediate these changes).

Correia-Martins, A., Tremblay, R., Bec, B., Roques, C., Atteia, A., Gobet, A., ... & Lagarde, F. (2022). Failure of bivalve foundation species recruitment related to trophic changes during an extreme heatwave event. *Marine Ecology Progress Series*, 691, 69-82.

Leung, J. Y., Russell, B. D., & Connell, S. D. (2019). Adaptive responses of marine gastropods to heatwaves. *One Earth*, 1(3), 374-381.

Spiecker, B. J., & Menge, B. A. (2022). El Niño and marine heatwaves: Ecological impacts on Oregon rocky intertidal kelp communities at local to regional scales. *Ecological Monographs*, 92(2), e1504.

Whalen, M. A., Starko, S., Lindstrom, S. C., & Martone, P. T. (2023). Heatwave restructures marine intertidal communities across a stress gradient. *Ecology*, 104(5), e4027.

Methods

Lines 296-297: Authors should put a parenthetical of landing weights or market value so the readers can better gauge what "sizeable" means in this context.

Figures: It would be good to have a colorscale key in the figures relating color to actual temperature. Currently, readers cannot gauge if a shift in shading represents a 0.1 °C change or a 5 °C change.

Line 315-316: The authors should add a sentence here explaining that clam holding conditions are similar to how clam diggers normally maintain their clams when conducting their fishing activities. Better yet would be to provide an estimate of how common this practice of holding sublegal clams is rather than immediately throwing them back when found.

Line 318-319: How similar was the fishing time during the heatwave to the other fishing times? To convince readers the change in clam burrowing was not due to stress from being in the bucket an extended period, it would be good to provide a table of the five fishing times in the supplemental material. Beyond changing the water once every 45-60 min was there any effort to keep bucket temperatures across the 5 experiments consistent, like keeping the buckets in shade? It does not appear so which may partially be the point of the experiment, but it would have been good to have a treatment where water quality in the buckets was kept consistent across experiment blocks while the other treatment was left to ambient conditions. Was water temperature or DO within the buckets ever measured? This experiment may have artificially stressed clams further than fishing activity since clam diggers are unlikely to spend an extra 30-60 min tethering clams in the air under the stressful conditions of a heatwave. This should at least be acknowledged in discussion if the authors cannot adequately describe how this was controlled for

Line 356: Here and throughout the manuscript, the authors should replace the term "experiment" with another word such as "block" or "trial" to improve clarity. It's all one experiment.

Line 392: Misspelled "August"

Lines 391-398: A notable experimental design issue is that the effects of temperature are only indirectly tested through temporal comparison. Investigating the effects of major environmental events is a challenge. The timing of these events is unpredictable, they are relatively rare, and events are hard to impossible to replicate making it logistically difficult to design good experiments around them. Before-After Control-Impact (BACI) designs are common when testing for the effects of such major events where impact sites and control sites are investigated near simultaneously, or when control sites are not able to be acquired, designing the experiment so that there is an amelioration of stress either by systematically increasing the distance from the source of impact or some other manipulation. This current study lacks control sites or experimental manipulation offsetting the temperature effects for a contrast. There are only measurements taken before and after the heatwave at the same site to serve as contrast and there is only one heatwave event measured in this time series. Furthermore, heatwave physical properties are not directly tested but instead blocked as different categorical periods of time that occur within the same site (i.e. "experiment" in the model) which can also have all sorts of other unknown seasonal or incidental environmental effects influencing the response. While it is too late for the authors to modify their experimental design, the authors might consider modifying their model to more directly test for the physical effects of heatwaves substituting temperature or humidex values in place of the month categories in the "experiment" variable and having time measured as a random effect. It is also possible that the relationship between the response variables and these physical values is not linear, but defined by crossing a threshold, in which case an additive instead of a linear model may be more appropriate.

Line 406: In the document providing R code, authors should add additional annotation so that readers can better follow the logic of the values chosen to produce model priors. In particular, why were variances of 10 chosen? Variances of 9 or standard deviations of 3 seem fairly common among users and variances of 6.25 (or 2.5 for standard deviations) is often the

default prior value for non-intercept parameters. Justification for the author's choices would be beneficial. Similarly provide justification for the matrix/diagonal dimensions (60 and 120) for the reader, which are the number of terms in their models.

Additionally, in the Supplementary Analysis document, it's stated that a variance of 8 was used, but in the code provided, it appears to be 6. Which value was actually used?

Results

Fig 1: It would be good to put in the caption how many measurements make up these averages.

Table S1: For reborrowing, only the intercept and "experiment" parameters had significant p-values, yet there were other model parameters that were still bolded. Bold graphics should be removed or explanation for the graphic should be provided in table caption.

Lines 124-125: This does not appear to be marginally non-significant. $p = 0.0123$ is significant.

Lines 124-129: Break this sentence up more cleanly.

Discussion

Lines 238-241: This wording implies that the change in predator abundance is due to the heatwave itself when the shift may also have a seasonal component. It is common for carnivorous/scavenging snails to become more abundant as spring shifts to summer, temperatures warm and salinity increases from reduced rainfall and river flow. The heatwave examined overlaps with the timing of some seasonal shifts. The authors results and discussion would benefit from some ruling out and acknowledging alternative hypotheses since the study site is near the mouth of a river which are prone to strong seasonal shifts. There is nothing about the authors measuring salinity and the weather station does not measure water quality parameters, but were rainfall patterns from the station relatively consistent?

Reviewer #2

(Remarks to the Author)

Dear Editor,

The submission COMMSBIO-24-6984-T concerns a field experiment that examined how mortality of soft shell clams varied between predator exclosure and control treatments in an estuary in Atlantic Canada. The 48-hour experiment was repeated five consecutive months. During one of these months (June), a heat wave occurred. In each experiment, the investigators fished for soft clams and handled them just as a recreational and commercial clammer throughout a multi-hour low tide. Of the harvest clams, sub-legal sized clams were then tethered and randomly assigned to plots (exclosure and control) at three different tidal elevation (low, mid, high). Throughout the experiment, the researchers quantified number of clams that reburied themselves as well as clam mortality, and predators. I enjoyed reading this well-written paper.

One suggestion for the authors to consider is focusing more of the Introduction and Discussion simply on how the by-product effects of harvesting may be worsened during heat waves. Relative to the paper's current focus on ecosystem function and species interactions during heatwaves, I believe that a focus on harvesting and by-catch is a simpler description of the dynamic that was explored by the study. This would also allow the authors to make connections to other published research about by-catch impacts or fishing-induced mortality during heat waves.

Another aspect of the study that concerned me is that the exclosure treatment did not fully prevent predation. Consequently, the only instance that promoted distinguish between environmentally-induced mortality and predator-induced mortality was the one experiment during the heatwave (June) after 24 hours. At that point, approximately 50% of clam mortality was due to environmental stress and 50% was due to predation. But other than that first 24 hour period during June, it was not possible to discern how much of the mortality was due to environment, predation, and scavenging. And if a significant amount of the predation was actually scavenging, then the cause of mortality is primarily fishing x environmental stress, not predation, which would contrast the main focus of the paper (heatwave and harvesting alters predator-prey interaction).

Finally, it is unclear whether the handling of the clams during the set-up of the experiment created more, less, or similar amounts of stress than the recreational and commercial clamming activities. To place the results of this study in context, it would be nice to have survey results of sub-legal size clam mortality on flats throughout the area during the heatwave and non-heatwave months. Or, alternatively, if the researchers could present some data on how the set-up of the experiment did not stress the clams during the heatwave.

Version 1:

Reviewer comments:

Reviewer #1

(Remarks to the Author)

The authors have substantially improved the quality of the manuscript, and I am pleased by how thoroughly the authors responded to my earlier comments. I only have a few minor revisions listed below.

Introduction

Lines 86-87: Consider rephrasing to “particularly if prey predator avoidance behaviours decrease²⁴”. While decreased activity levels in many bivalves would make them more vulnerable to predators, decreased activity levels in general does not always increase vulnerability. Many predator avoidance behaviors instead rely on decreasing activity level to avoid predation (e.g. crabs reducing activity in the presence of fish or freezing behavior in mammals).

Methods

Figure S4: Should add color scale to figure.

Line 400-403: The sentence structure here is a little confusing with how the parenthetical was embedded. Were holding conditions also designed to minimize physiological stress associated with air exposure? Or is this something else that not all clam diggers do? If it's the latter, add it to the parenthetical. If it's the former, either move the parenthetical to the end of the sentence or change the second half of the sentence starting with “as well as” into its own new sentence (e.g. Holding conditions were also designed to minimize...).

Line 419: Here might be a good spot to mention that this time period was well within the amount of time diggers typically spend harvesting clams and then provide a normal range of time diggers are out harvesting. (Although it is mentioned in the discussion).

Line 503: “trails” should be “trials”

Line 513: Listing the three categorical variables in the model as a single 3-way interaction in the model description seems a little unconventional. Did you only include the 3-way interaction or did you also include the lower order 2-way interactions? If you did include them, at least make “interactions” plural in the sentence.

Lines 521-522: What is meant by “fishes the traps approximately”? Seems like an incomplete phrase on how frequently traps were fished.

Figure S1 caption: The note about the May trial not being represented is stated twice.

Results

Line 115: “experiment al trials” should be “experimental trials”.

Line 116: change “The May experiment” to “The May trial”

Lines 184-185: Need to finish the phrase “which is consistent [with the BGLMM results]”.

Discussion

Line 254: rephrase to “do not add water, but just keep the clams in a dry bucket”

Lines 260-262: The information in this sentence is repetitive of the information provided in lines 252-254. Should integrate this information there.

Reviewer #2

(Remarks to the Author)
Dear Editor,

I reviewed the original submission by Clements et al and have now reviewed the revised submission (COMMSBIO-24-6984A). I am concerned about the authors' inferences throughout the paper given the study design and results. In my initial read of the resubmission, I strongly disagreed with most of the author's conclusions about the role of predation because there was so much clam mortality in the predator enclosures (presumably due to physiological stress, not predation) and because the purported predator species (crab and snail) are well-known scavengers of dead things. Furthermore, while some predators may increase their foraging activity under elevated temperature, it's also well appreciated that more dead things lead to more scavenger activity. Consequently, a large abundance of dead clams could explain the higher activity of crabs and snails. At this point, I felt that the more conservative interpretation of the results was that heat and fishing interactively caused clam mortality, with a trophic benefit to scavengers.

But then I went for a walk and thought further about this paper. I revisited the results (specifically, Figure 2) and noted that the clam mortality after 24 hours in the predator enclosure treatment was ~35%. In contrast, the mortality of clams after 24 hours in the predator inclusion treatment was ~70%. Unfortunately, the predator enclosure treatment did not fully exclude all

predators. But if we conservatively assume that all dead clams in the predator enclosure treatment were killed solely by heat and physiological stress, then I inferred that half of the mortality of clams in the predator inclusion treatment could be allocated to predation and the other ½ would be allocated to physiological stress (promoting scavenging). Furthermore, because the clam mortality after 48 hours was equally high in the predator enclosure and inclusion treatments, this enhanced predation interaction would be rather ephemeral, quickly converting from predation to scavenging as day 1 (24 hours) proceeded into day 2 (48 hours). This inference decreased my overall concern.

However, I then revisited the authors' rebuttal letter and the revised submission with a focus on paragraph five of the Discussion. The rationale and logical argument of this paragraph did not alleviate my concern. I suggest that the authors revise this paragraph to walk the reader through the situation I outlined above, which will highlight a potential role for predation would be equal in strength to stress-induced mortality, but of much shorter duration.

Another observation is that fourth paragraph of the Discussion about clam digging methods is nice, but it seems overkill. I suggest trimming it down. In addition, in paragraph 3 of the Discussion, the central argument is that increasing temperature enhances predator activity and that is what happened in the current study. But there is zero mention of a simpler explanation....heat waves x human disturbance can increase mortality of clams. Lots of dead clams attract more scavengers. At least revise this paragraph to give this simpler explanation an equal chance. Finally, the abstract is well written, but it opens so generally that a reader could envision a central topic of study to be that humans are using their air conditioning during a heat wave or over using freshwater etc. I suggest revising the opening sentence so that it is a less vague.

Version 2:

Reviewer comments:

Reviewer #2

(Remarks to the Author)

The authors addressed my previous comments and concerns.

Note below that our general responses are in *blue text*, and the specific text additions/changes are in *red text*. Line numbers refer to the CLEAN manuscript.

Reviewer #1 (Remarks to the Author):

In the manuscript “Human activity during an extreme heatwave alters predator-prey dynamics in a ubiquitous nearshore system”, the authors conduct a field experiment to investigate how clam fishing activity can interact with marine heatwaves to reduce the burrowing behavior of released sublegal sized clams and increase their mortality rates. The effects of marine heatwaves have been an increasingly hot topic these past few years, and the findings from this experiment are dramatic and pertinent to a wide audience. Field experiments on the interactive effects of human activity and extreme events are relatively rare and difficult to conduct rigorously. The biggest issue with this study is that there is not a good control for the heatwave either by experimental manipulation of surroundings or use of alternative sites simultaneously. There is only one heatwave examined which overlaps with some seasonal timing and all before and after measurements occur at only one site which opens up the possibility of confounding factors. Given the magnitude of differences between heatwave and non-heatwave observations, most of the observed variation is likely to be attributed to the heatwave. However, it would be good for the authors to spend more effort refuting alternative hypotheses and more directly testing for the effects of the heatwave. Other issues are that the work needs to be better placed in the context of the broader literature as well as more thoroughly discuss/describe how their methods relate to and depart from standard clam fishing methods. Overall, this paper is well-written, interesting, and will be a good contribution to the literature.

- **Response:** Thanks to the reviewer for the positive and constructive assessment of our manuscript. In the revised manuscript, we have used the reviewer’s suggestions to address concerns related to temporal confounding factors (e.g. seasonality and other acute environmental factors), including additional analyses (GAMM) and additional data independent of our experimental work. We have also expanded our Discussion to address concerns related to clam digging vs. experimental handling of clams, and use the references provided by the reviewer (as well as others) to better place to work in the context of existing literature, taking care to reduce exaggeration in the Introduction.

Abstract

1. Lines 18-19: The wording on the second half of this sentence needs to be toned down. “Absent” is an exaggeration. For instance, Freedman et al. (2020) found that marine heatwaves had similar effects for kelp forest fish communities regardless of regional fishing activity while Zhang et al.’s (2022) data indicate heatwaves intensify the effects of nutrient loading and herbicides.

Freedman, R. M., Brown, J. A., Caldow, C., & Caselle, J. E. (2020). Marine protected areas do not prevent marine heatwave-induced fish community structure changes in a temperate transition zone. *Scientific Reports*, 10(1), 21081.

Zhang, P., Wang, T., Zhang, H., Wang, H., Hilt, S., Shi, P., ... & Xu, J. (2022). Heat waves rather than continuous warming exacerbate impacts of nutrient loading and herbicides on aquatic ecosystems. *Environment International*, 168, 107478.

- **Response:** Thanks to the reviewer for the references both here and in the first comment of the Introduction (below). We have revised the Introduction accordingly (with additional references) and also revised the text in the Abstract on Lines 18-19, which now reads:

Heatwaves may have multifaceted ecological impacts; however, field studies assessing the ecological effects of human activity during heatwaves are rare.

2. Line 22-25: This sentence needs to be split in two.

- **Response:** The text has been amended on Lines 21-23 and now reads:

During monthly field trials from May-September 2024, we simulated clam fishing at low tide and tracked the reburrowing and mortality rates of marked, sub-legal sized clams returned to the sediment surface. Half of the returned clams were protected from crab predation and estimates of predator activity were recorded.

Introduction

3. Lines 60-61: The authors need to be careful not to exaggerate. There is a lot of work showing how nearshore coastal systems respond to marine heatwaves with and without mass mortality (e.g. Spiecker and Menge 2022, Whalen et al. 2023). Theoretical work on this subject is also common and lessons can be derived from other disasters that produce similar effects as marine heatwaves to estimate how ecological interactions can change. There are even a few studies that explicitly look at how ecological interactions shift in sessile nearshore systems due to marine heatwaves (e.g Correia-Martins et al. 2022). The available literature widens when non-sessile organisms are included (e.g. Leung et al. 2019). The authors should instead focus on the fact that such field studies are rare and that they are necessary to identify the underlying mechanisms for how populations and communities will shift (to better predict and remediate these changes).

Correia-Martins, A., Tremblay, R., Bec, B., Roques, C., Atteia, A., Gobet, A., ... & Lagarde, F. (2022). Failure of bivalve foundation species recruitment related to trophic changes during an extreme heatwave event. *Marine Ecology Progress Series*, 691, 69-82.

Leung, J. Y., Russell, B. D., & Connell, S. D. (2019). Adaptive responses of marine gastropods to heatwaves. *One Earth*, 1(3), 374-381.

Spiecker, B. J., & Menge, B. A. (2022). El Niño and marine heatwaves: Ecological impacts on Oregon rocky intertidal kelp communities at local to regional scales. *Ecological Monographs*, 92(2), e1504.

Whalen, M. A., Starko, S., Lindstrom, S. C., & Martone, P. T. (2023). Heatwave restructures marine intertidal communities across a stress gradient. *Ecology*, 104(5), e4027.

- **Response:** Thanks again to the reviewer for this suggestion and for pointing us to relevant references. We have now toned down the exaggerated text in the Introduction as suggested and have incorporated the references provided by the reviewer into the text (along with one other reference to a paper that was published in December; *Jawad et al 2024, JEMBE, 581: 152060*). Please note that we also revised the text here based on comments from Reviewer 2 to refocus

the Intro on fisheries rather than ecology (while we elected to not refocus completely on fisheries, we did add some additional fisheries context to the Intro here). The relevant text in the Introduction on Lines 57-72 now reads:

Consequently, contemporary heatwaves have led to mass mortalities of sessile nearshore organisms^{4,18,19}. Even when mass mortalities do not occur, studies suggest that heatwaves can result in broad ecological changes in nearshore systems²⁰⁻²⁴.

While studies suggest that heatwaves can have ecological effects, adequate in situ field studies in nearshore systems are rare²⁵. Such field studies are necessary to identify ecological mechanisms that may drive population- and community-level changes imposed by climate change, and to predict and adapt to these changes. Moreover, although some studies exist^{26,27}, research on the potential ecological effects of anthropogenic disturbances during heatwaves is limited. The act of fishing, for example, can result in ecological shifts by reducing prey abundances through the removal of targeted and non-targeted (bycatch) species²⁸. While this knowledge has underpinned calls for ecosystem based fisheries management since the 1990s²⁸, the ways in which heatwaves might influence the ecosystem effects of fishing remains understudied for many fisheries. As such, an understanding of how fishing during heatwaves influences predator-prey dynamics and fishing mortality is paramount for developing effective ecosystem based fisheries management as the climate continues to change^{28,29}.

One globally-common nearshore fishery that has the potential to alter ecological function is clam digging. Clam digging during low tide can be destructive, as it...

Methods

4. Lines 296-297: Authors should put a parenthetical of landing weights or market value so the readers can better gauge what “sizeable” means in this context.

- **Response:** We added 2023 landing weights and market value for the southern Gulf of St. Lawrence commercial soft-shell clam fishery as suggested. For broad context regarding the US and Canadian fisheries, we refer the reader to a recently-published book chapter on this topic: Kennedy, V. S. History of fisheries for the soft-shell clam *Mya arenaria*. in *The Soft-Shell Clam Mya arenaria: Biology, Fisheries, and Mariculture* (eds. Kennedy, V. S., & Beal, B. F.) 423-472 (American Fisheries Society, 2023). The text on Lines 346-349 now reads:

This species constitutes a sizable recreational and commercial fisheries in eastern North America (Canada and the USA)⁵⁹, including the region where this study took place (2023 southern Gulf of St. Lawrence commercial soft-shell clam landings = 378,965 metric tons valued at CA\$1,557,383; Clements, under review).

5. Figures: It would be good to have a colorscale key in the figures relating color to actual temperature. Currently, readers cannot gauge if a shift in shading represents a 0.1 °C change or a 5 °C change.

- **Response:** We added colour gradient legends to Figures 2 and 3. Gradient legends are not applicable to Figure 4 and we feel are not necessary in Figure 1 because the temperature values

are on the y-axis. We have also changed the x-axis labels to read “Trial” instead of “Experiment” in Figures 1-3.

6. Line 315-316: The authors should add a sentence here explaining that clam holding conditions are similar to how clam diggers normally maintain their clams when conducting their fishing activities. Better yet would be to provide an estimate of how common this practice of holding sublegal clams is rather than immediately throwing them back when found.

- **Response:** Unfortunately, there is no way to really determine just how common this holding method is among clam diggers, as methods will vary across individual clam diggers and there is no reporting of holding conditions from clam diggers. Nonetheless, we added a sentence justifying why we chose these particular holding methods to the Methods on Lines 366-369, which reads:

Holding conditions were aimed to mimic methods used by many clam diggers (based on personal conversations with a few clam diggers prior to experimentation; although not all clam diggers fill their buckets with water), as well as minimize physiological stress associated with air exposure while the clams are out of their burrows.

7. Line 318-319: How similar was the fishing time during the heatwave to the other fishing times? To convince readers the change in clam burrowing was not due to stress from being in the bucket an extended period, it would be good to provide a table of the five fishing times in the supplemental material. Beyond changing the water once every 45-60 min was there any effort to keep bucket temperatures across the 5 experiments consistent, like keeping the buckets in shade? It does not appear so which may partially be the point of the experiment, but it would have been good to have a treatment where water quality in the buckets was kept consistent across experiment blocks while the other treatment was left to ambient conditions. Was water temperature or DO within the buckets ever measured? This experiment may have artificially stressed clams further than fishing activity since clam diggers are unlikely to spend an extra 30-60 min tethering clams in the air under the stressful conditions of a heatwave. This should at least be acknowledged in discussion if the authors cannot adequately describe how this was controlled for.

- **Response:** This is a good point. While we did not keep track of detailed fishing time, we do have information on the time that fishing started and the time that the clams were released into the experimental plots. We now include a table of this in the Supplementary Material (**Table S7**). Notably, the time from the start of fishing to the start of the experiment during the June experiment did not deviate from the other experiments. In fact, this time between fishing and the onset of experiments was longer for the three latter experimental trials (July-September) than in June. As such, increased fishing or handling times during the heatwave cannot explain the results we observed. We did not make any attempts to keep bucket temperatures consistent. As the reviewer mentioned, this was partly the point of the experiment, as clam diggers would not have done this either. Unfortunately, we did not have loggers monitoring temperature or DO in the buckets (now explicitly mentioned in the Discussion on Lines 217-218), as the study did not set forth to test the effects of heatwaves from the get-go, but rather took advantage of a heatwave when one was predicted. Nonetheless, the reviewer makes a good point about the tethering in that clam diggers are unlikely to do this when they fish clams. However, it is important to note that not all clam diggers use the same methods when fishing for clams, as

there are no standardized management rules for how to handle/hold clams while fishing. While our methods do not replicate all of the ways in which clam diggers fish for clams, we do know that clam diggers will fish for the maximum possible duration (typically 5-6 hours) that they can during low tide, particularly in the Gulf of St. Lawrence because tide cycles are semi-diurnal with very small tidal ranges (and, as such, days where clam diggers can fish are limited throughout the year). Given this, it is unlikely that our fishing methodology and the exposure of clams to air would invoke more stress on the clams than would be invoked by many clam diggers. Nonetheless, we do recognize that it is possible that some fishing methods may reduce physiological stress during heatwaves, and that further experiments determining whether particular methods during fishing can alleviate the mortality observed in our study are warranted. We have added a full paragraph in the Discussion to incorporate these limitations and suggestions for future study on Lines 228-252:

It is important to note that while our experiment attempted to mimic typical clam digging methods in the region, some aspects of our experiment deviated from what clam diggers would do. Notably, when digging for clams, clam diggers would not tether washers to their sub-legal catch as we did in our experiment. Unfortunately, clam digging method is something that is difficult to control for because not all clam diggers use the same methods when fishing for clams, as there are no standardized management rules for how to handle/hold clams while fishing. For example, some clam diggers will put the clams in a bucket of water while other diggers don't add water, they just keep the clams in a dry bucket. If using water, some clam diggers would change out this water regularly as we did while others would not. Some clam diggers will measure and toss clams back on the fly while others will wait until they are finished fishing. While our methods do not replicate all of the ways in which clam diggers fish for clams, clam diggers most often fish for the maximum possible duration that they can during low tide (typically 5-6 hours), particularly in the Gulf of St. Lawrence because tide cycles are semi-diurnal with small tidal ranges (days where clam diggers can fish are limited throughout the year). Furthermore, not all clam diggers will attempt to minimize stress on their catch by placing clams in water, as many diggers simply place their catch in an empty bucket because filling it with water makes it heavy and would increase physical fatigue. Anecdotally, during one of the trials, we observed a recreational clam digger fish for clams for a longer duration than our fishing and tethering process, place all clams in a bucket without water, and measure and toss back the sub-legals once they were finished digging. Such an approach during a heatwave would expose the fished clams to both desiccation and extreme air temperatures, likely maximizing physiological stress. As such, it is unlikely that our fishing methodology and the exposure of clams to air would invoke more stress on the clams than would be invoked by the practices of many clam diggers. Nonetheless, there may exist some optimal clam digging method that could reduce stress on the animals when fishing during heatwaves or other acute environmental stresses. Further studies exploring the influence of different holding conditions when clam digging during periods of environmental stress on sub-legal burrowing and survival are thus warranted.

8. Line 356: Here and throughout the manuscript, the authors should replace the term “experiment” with another word such as “block” or “trial” to improve clarity. It’s all one experiment.

- **Response:** Amended. We have replaced the term ‘experiment’ with ‘experimental trial’, or simply ‘trial’ throughout the manuscript. We also amended this in the figures as well.

9. Line 392: Misspelled “August”

- **Response:** Fixed. Text on Lines 442-443 now reads:
*...independent and interactive effects of experimental trial (fixed categorical factor with 5 levels: May, June, July, **August**, September)...*

10. Lines 391-398: A notable experimental design issue is that the effects of temperature are only indirectly tested through temporal comparison. Investigating the effects of major environmental events is a challenge. The timing of these events is unpredictable, they are relatively rare, and events are hard to impossible to replicate making it logistically difficult to design good experiments around them. Before-After Control-Impact (BACI) designs are common when testing for the effects of such major events where impact sites and control sites are investigated near simultaneously, or when control sites are not able to be acquired, designing the experiment so that there is an amelioration of stress either by systematically increasing the distance from the source of impact or some other manipulation. This current study lacks control sites or experimental manipulation offsetting the temperature effects for a contrast. There are only measurements taken before and after the heatwave at the same site to serve as contrast and there is only one heatwave event measured in this time series. Furthermore, heatwave physical properties are not directly tested but instead blocked as different categorical periods of time that occur within the same site (i.e. “experiment” in the model) which can also have all sorts of other unknown seasonal or incidental environmental effects influencing the response. While it is too late for the authors to modify their experimental design, the authors might consider modifying their model to more directly test for the physical effects of heatwaves substituting temperature or humidex values in place of the month categories in the “experiment” variable and having time measured as a random effect. It is also possible that the relationship between the response variables and these physical values is not linear, but defined by crossing a threshold, in which case an additive instead of a linear model may be more appropriate.

- **Response:** We want to thank the reviewer for their analytical suggestions here, which are very helpful. While we do not want to negate or alter the original analyses that we set forth (in the interest of analytical transparency, and because we feel that the analysis and interpretation of Figure 2 is easily interpretable and digestible for the broad readership of the journal), we do agree that incorporating average air temperature and testing that effect directly is beneficial. Indeed, this is what we were getting at with the temperature response curves in Figure 4, which indeed suggests that the relationship between air temperature and burrowing and mortality is not necessarily linear. As such, we have elected to keep our original BGLMM analysis in the paper to interpret Figure 2 data, but have added generalize additive mixed modelling to more directly test for the effects of air temperature as suggested by the reviewer. Specifically, we built two negative binomial GAMMs (one for burrowing and one for mortality) to test for the effects of average air temperature (fixed continuous smooth term), predator treatment x time since fishing x tide level as interactive fixed categorical parametric terms, and Julian date as a random continuous smooth term. The results of this analysis aligned very well with those of the BGLMM analysis and suggest that air temperature had a highly significant effect on both burrowing and mortality. Additionally, there were no significant effects of the parametric terms for burrowing, and a significant predator treatment x time since fishing interaction for the mortality, which is

exactly what was observed in the BGLMM analysis, which speaks to the robustness of our results. We have added descriptions of the results and methods for this additional analysis to the manuscript on Lines 164-168 (Results) and 468-480 (Methods) and include full statistical results in a new supplementary table (**Table S6**). The corresponding Results and Methods read:

RESULTS: *We also used generalized additive mixed modelling to directly test for effects of air temperature during fishing, predator treatment, tide level, and trial on clam reburrowing and mortality; models included Julian date as a random factor. The GAMM models revealed highly significant effects of air temperature on clam reburrowing and mortality (Table S6). There were no significant effects of any other fixed factors on reburrowing, which is consistent (although there was a significant effect of Julian date; Table S6). Similarly, the GAMM model results for the effects of other fixed factors on clam mortality were consistent with the BGLMM model results. Herein, there was a significant interactive effect of predator treatment × time since fishing, as well as significant independent effects of those two factors (Table S6).*

METHODS: *It is important to note that evidence for a heatwave effect from the BGLMMs above is based on categorical time periods for experimental trials conducted at the same site. As such, evidence for heatwave effects on clam burrowing and mortality are restricted to temporal correlations between observed biotic responses and temperature conditions during those categorical timepoints. However, other environmental influences not related to temperature that vary over time could potentially influence any observed biotic responses. For example, low salinity events are known to drastically reduce soft-shell clam burrowing rates²¹. Similarly, seasonal shifts in predator and prey activity could also influence the observed results. To account for this, we supplemented the BGLMM approach above with additional analyses to more directly test for the effects of air temperature during fishing on clam reburrowing and mortality rates. Specifically, we built negative binomial generalized additive mixed models (one for reburrowing and one for mortality) with average air temperature during fishing as a continuous smooth term, the interaction between predator treatment × time since fishing × tide level as categorical parametric terms, and Julian date as a continuous smooth random term.*

In addition to the GAMMs, we also provide additional supplemental data (**Figs S1-S3**) that argue against seasonal/other environmental explanations for the observed trends in predator activity and burrowing/mortality, as suggested both here and in another comment below. First, while we do not have comparative data for mudsnails, we utilize data from a nearby site of a crab monitoring program in Kouchibouguac NP, where crab counts in modified Fukui traps were obtained ~weekly from June-October from 2019-2024 (**Fig S1**). These data show that the increased numbers of crabs in our predator inclusion plots are likely not due to a seasonal increase in crab activity in June. Herein, while the 2024 monitoring data follow the same general trend that we observed in our experiments, no other year in the dataset (2019-2023) displayed a similar increase in crab abundance during the month of June. In fact, the June 2024 crab catches in the monitoring program were 2X higher than the crab catch in next highest year for the same time period. This strongly suggests that the heatwave was indeed the cause of the increased predator activity observed in our experiment. Second, we re-purposed existing data from a

previous study from our lab group (Ledoux et al. 2023) where we recorded sub-legal soft-shell clam reborrowing rates in predator exclusion cages at four different sites in Kouchibouguac NP (including the same study site in this paper) during the summer of 2021, where a heatwave did not occur. These data show that clam reborrowing rates do not seasonally drop during the same time period in June, as the percentage of reborrowed clams 24h after being fished on June 21, 2021 was 85.6% on average, compared to 8.7% 24h after being fished on June 20, 2024 in this study (**Fig S2**). However, that previous study did suggest that low salinity events due to heavy rainfall can drastically reduce reborrowing rates. While we did deploy a salinity logger at our study site during the experiments, the logger unfortunately malfunctioned and did not record any data. As such, and at the suggestion of the reviewer, we provide daily precipitation data obtained from the Kouchibouguac Weather Station (same source as the air temperature data) for each of the five months in which experimental trials were conducted to show that there was no rainfall that occurred before or during the June experiment that would have driven low salinity conditions at the study site (**Fig. S3**).

Ultimately, these additional data provide strong evidence that it was indeed the impact of fishing during a heatwave that drove the observed burrowing/ mortality results in this study, particularly given that successful reborrowing would confer high rates of survival after being fished. Again, we want to thank the reviewer for their suggestions here—they truly make for a much more compelling case that the heatwave was causative.

We have now added details on data sources and methodology for the additional data outlined above (Lines 482-524), which reads:

Exploring alternative explanations for differences between trials

*To investigate potential seasonal effects on the observed trends in predator activity across the experimental trials, we leveraged weekly data from a crab monitoring program conducted near our study site by Kouchibouguac National Park from 2019 - 2024 (independent from the data in our experiment). This annual program, designed to monitor invasive European green crabs (*Carcinus maenas*) in the park⁶⁷, deploys a series of modified Fukui traps at various sites throughout Kouchibouguac National Park and fishes the traps approximately, recording counts of all crab species found in the traps, as well as the number of days the traps were deployed. One particular site (named 'Loggiecraft') located near our experimental study site has been monitored consistently across years, and we were able to obtain data on crab catches and fishing effort back to 2019. We thus plotted average crab catch per day (total number of crabs ÷ number of days traps were deployed; average of n = 2-4 traps) recorded on each of the fishing days to observe temporal trends in crab catches from this monitoring program over the past six years (Fig. S1). We used these trends to descriptively compare those to the estimates of crab activity in our experimental trials. If crab counts in our experimental plots were seasonally driven, we would expect to see similar temporal trends in crab catches in the monitoring data as we did in our experimental plots (i.e., increasing crab catches from early to late June with a sharp decrease and stabilization thereafter) across years.*

To further investigate if the observed differences in burrowing and mortality between experimental trials could be attributed to seasonal cycles in clam burrowing rather than the June heatwave, we leveraged existing unpublished data from our lab group that is complementary to, but not included in, the results reported in 30. In that study, we measured sub-legal soft-shell clam reburrowing times in predator exclusion cages at four different sites in Kouchibouguac National Park (including the same study site in this paper) during the summer of 2021, where a heatwave did not occur. Incidentally, this provided us with additional data on sub-legal clam reburrowing rates 24 hours after being fished from June 21, 2021 (no heatwave) and four time points thereafter to which we could compare our 2024 experiment data to. While differences in experimental design preclude us from comparing the burrowing frequencies statistically (the 2021 study had multiple sites and only deployed three clams per plot), we nonetheless used the data to visually assess temporal trends in sub-legal clam reburrowing across two different years (Fig. S2). Herein, if trends in burrowing activity were seasonally driven, we would expect to observe a similar reduction in the percentage of clams reburrowed in the June trials for both years.

Acute changes in other environmental parameters, such as decreases in salinity following heavy rainfall events, have the potential to reduce sub-legal soft-shell clam reburrowing rates³⁰ and could thus drive observed differences between experimental trials. While we did deploy a salinity logger (Star-Oddi) at the study site over the duration of the experimental trials, the logger unfortunately malfunctioned and did not record any salinity data. In lieu of in situ salinity conditions, we retrieved daily precipitation data from the Kouchibouguac weather station (same source as the air temperature data) for the duration of the experiments to determine if any heavy rainfall events prior to the trial could have resulted in low salinity conditions during the trial. We then plotted daily precipitation totals for each of the five months of the experimental trials (May-September; Fig. S3). If the anomalous burrowing/mortality and predator activity estimates during the June trial in our experiment were driven by low salinity during the trial, we would expect to observe heavy rainfall event(s) in the day(s) leading up to the June trial, as well as a stark absence of heavy rainfall events leading up to the other experimental trials (where proportions of reburrowing were high and mortality low).

We have also added a full paragraph to the Discussion to put these additional data into context and rule out other potential explanations for the reduced burrowing, increased mortality, and increased presence of predators/scavengers in the June trial (Lines 283-305), which reads:

One potential pitfall of our study is that our experiment was conducted at a single site and replicated biological observations through time over five separate experimental trials. As such, our design lacks a comparative control treatment with experimental trials conducted at the same time in the absence of a heatwave. Indeed, this is perhaps one of the reasons why in situ climate change experiments are rare, as deploying a suitable control treatment during climate events such as heatwaves is highly challenging. As such, it is possible that the observed changes in the June experiment were driven by factors other than the heatwave. For example, predictable seasonal shifts in predator-prey activity as Spring transitions to Summer may have driven the reduced burrowing and increased predator activity in June.

With respect to predator activity, data spanning 2019-2024 from a crab monitoring program revealed no predictable increase in predator activity between late May and early June as observed in our experiment (Fig. S1). Interestingly, 2024 data from the crab monitoring program followed a similar trend in crab catches as the estimates of predator activity recorded in our experimental plots, with crab catches for the month of June being 2× higher in 2024 than any of the five preceding years. Together, our data and that of the crab monitoring program strongly suggest that the observed increase in predator activity in the June trial of our experiment were indeed driven by the heatwave. Similarly, comparing the reborrowing rates of sub-legal clams at the same site between two years revealed that reborrowing rates do not predictably plummet during similar dates June, but rather appear to be influenced by acute environmental changes such as heatwaves and low salinity events (Fig. S2). However, we did not observe any evidence that an acute salinity change could have driven the observations in our June trial, as there was no rainfall observed nine days before, nor during, the June experimental trail (Fig. S3), and salinity effects only appear after extreme rainfall events (e.g., 70+ mm in <24h²⁷). Coupled with these additional data sources, our results strongly suggest a causal relationship between fishing clams during the extreme heatwave in June and the observed shifts in predator-prey activity, and increased sub-legal clam mortality.

11. Line 406: In the document providing R code, authors should add additional annotation so that readers can better follow the logic of the values chosen to produce model priors. In particular, why were variances of 10 chosen? Variances of 9 or standard deviations of 3 seem fairly common among users and variances of 6.25 (or 2.5 for standard deviations) is often the default prior value for non-intercept parameters. Justification for the author's choices would be beneficial. Similarly provide justification for the matrix/diagonal dimensions (60 and 120) for the reader, which are the number of terms in their models.

- **Response:** Firstly, we want to thank the reviewer for taking the time and attention to assess our data and code; we very much appreciate this. The first author of the paper has published over 20 papers with open data and code, and this is the first time a reviewer has taken the time to assess and comment on the code. We now provide the rationale for selecting a variance of 10 in the Methods (Lines 455-463) and in the R code itself. The additional text here reads:

A variance prior of 10 was selected due to the high degree of variance in the data driven by near-complete separation, and a diagonal matrix value of 60 was chosen to match the number of terms in the models. We first tried fitting the models with a variance prior of 9 (typical for datasets with near-complete separation); however, the PIRLS step-halvings for the mortality model failed to adequately reduce the variance in `pwrssUpdate`, suggesting that our variance prior did not capture the actual variance in the data. We therefore selected a slightly larger variance prior of 10, which alleviated the error, and applied the same variance prior to the burrowing model for model consistency. Notably, applying a variance of 9 or 10 for the burrowing model did not meaningfully alter model output, suggesting that results for the burrowing model are reasonable.

12. Additionally, in the Supplementary Analysis document, it's stated that a variance of 8 was used, but in the code provided, it appears to be 6. Which value was actually used?

- **Response:** We did indeed use a variance of 6 in the models as the variance in the data was not as extreme as the main dataset used. We have amended the supplementary analysis file as such and also added the justification for the diagonal matrix value there as well. The text in the supplementary analysis now reads:

Briefly, models were constructed using the `bglmer()` function the ('blme' package; Chung et al. 2013), including plot ID as a categorical random variable to account for random spatial effects and repeated measures across the eight time points, and specifying weak zero-mean normal priors (6,120) to account for near-complete separation (as per Bolker 2018). The lower variance prior was chosen due to the removal of predator treatment as a fixed factor and the increased number of replicates for each fixed factor combination level; the diagonal matrix value of 120 was chosen to match the number of terms in the models.

Results

13. Fig 1: It would be good to put in the caption how many measurements make up these averages.

- **Response:** Amended. Number of measurements for the data points in each panel have been added to the figure caption, which reads:

n = 24 measurements (1 per hour × 24 hours) for each point in top and middle panels; n = 4 measurements (1 per hour × 4 hours) for each point in bottom panel.

14. Table S1: For reborrowing, only the intercept and “experiment” parameters had significant p-values, yet there were other model parameters that were still bolded. Bold graphics should be removed or explanation for the graphic should be provided in table caption.

- **Response:** Good catch! We actually noticed this after submission and revised the supplementary material accordingly. This has now been amended and we have double checked all supplemental tables to make sure there are no errors in bolding.

15. Lines 124-125: This does not appear to be marginally non-significant. $p = 0.0123$ is significant.

- **Response:** The text on Lines 122-124 has been amended to state a significant effect, not a marginally non-significant one, and now reads:

With respect to clam mortality, the BGLMM model uncovered significant trial × time since fishing ($\chi^2_4 = 12.80$, $p = 0.0123$, Table S1) and predator treatment × time since fishing interactions ($\chi^2_1 = 11.53$, $p = 0.0007$, Table S1).

16. Lines 124-129: Break this sentence up more cleanly.

- **Response:** We have broken this into two sentences more cleanly as suggested. The text on Lines 122-126 now reads:

With respect to clam mortality, the BGLMM model uncovered significant trial × time since fishing ($\chi^2_4 = 12.80$, $p = 0.0123$, Table S1) and predator treatment × time since fishing interactions ($\chi^2_1 = 11.53$, $p = 0.0007$, Table S1). There was also a marginally non-significant three-way interactive effect between trial, predator treatment, and time since fishing ($\chi^2_4 = 8.40$, $p = 0.0781$, Table S1). We thus explored pairwise comparisons across relevant levels of all three factors.

Discussion

17. Lines 238-241: This wording implies that the change in predator abundance is due to the heatwave itself when the shift may also have a seasonal component. It is common for carnivorous/scavenging snails to become more abundant as spring shifts to summer, temperatures warm and salinity increases from reduced rainfall and river flow. The heatwave examined overlaps with the timing of some seasonal shifts. The authors results and discussion would benefit from some ruling out and acknowledging alternative hypotheses since the study site is near the mouth of a river which are prone to strong seasonal shifts. There is nothing about the authors measuring salinity and the weather station does not measure water quality parameters, but were rainfall patterns from the station relatively consistent?

- **Response:** As mentioned, we now include additional data and a full paragraph in the Discussion ruling out seasonal influence in predator activity and burrowing behaviour, as well as any influence of acute salinity change. The Discussion text on Lines 283-305 reads:

One potential pitfall of our study is that our experiment was conducted at a single site and replicated biological observations through time over five separate experimental trials. As such, our design lacks a comparative control treatment with experimental trials conducted at the same time in the absence of a heatwave. Indeed, this is perhaps one of the reasons why in situ climate change experiments are rare, as deploying a suitable control treatment during climate events such as heatwaves is highly challenging. As such, it is possible that the observed changes in the June experiment were driven by factors other than the heatwave. For example, predictable seasonal shifts in predator-prey activity as Spring transitions to Summer may have driven the reduced burrowing and increased predator activity in June. With respect to predator activity, data spanning 2019-2024 from a crab monitoring program revealed no predictable increase in predator activity between late May and early June as observed in our experiment (Fig. S1). Interestingly, 2024 data from the crab monitoring program followed a similar trend in crab catches as the estimates of predator activity recorded in our experimental plots, with crab catches for the month of June being 2× higher in 2024 than any of the five preceding years. Together, our data and that of the crab monitoring program strongly suggest that the observed increase in predator activity in the June trial of our experiment were indeed driven by the heatwave. Similarly, comparing the reborrowing rates of sub-legal clams at the same site between two years revealed that reborrowing rates do not predictably plummet during similar dates June, but rather appear to be influenced by acute environmental changes such as heatwaves and low salinity events (Fig. S2). However, we did not observe any evidence that an acute salinity change could have driven the observations in our June trial, as there was no rainfall observed nine days before, nor during, the June experimental trail (Fig. S3), and salinity effects only appear after extreme rainfall events (e.g., 70+ mm in <24h27). Coupled with these additional data sources, our results strongly suggest a causal relationship between fishing clams during the extreme heatwave in June and the observed shifts in predator-prey activity, and increased sub-legal clam mortality.

Reviewer #2 (Remarks to the Author):

General: The submission COMMSBIO-24-6984-T concerns a field experiment that examined how mortality of soft shell clams varied between predator enclosure and control treatments in a an estuary in Atlantic Canada. The 48-hour experiment was repeated five consecutive months. During one of these months (June), a heat wave occurred. In each experiment, the investigators fished for soft clams and handled them just as a recreational and commercial clambers throughout a multi-hour low tide. Of the harvest clams, sub-legal sized clams were then tethered and randomly assigned to plots (enclosure and control) at three different tidal elevation (low, mid, high). Throughout the experiment, the researchers quantified number of clams that reburied themselves as well as clam mortality, and predators. I enjoyed reading this well-written paper.

- **Response:** Thanks to the for their constructive comments on our manuscript. We are happy to hear that the reviewer enjoyed the read. We address comments from the reviewer below. In short, we have elected to keep an ecological focus, but have added text to the Introduction to more explicitly add fisheries context. Additionally, we have expanded our discussion to address concerns regarding predator exclusions/predation vs. scavenging vs. environmental (physiological) mortality, as well as to address concerns regarding our handling procedure vs. clam digging. Detailed responses are below.

1. One suggestion for the authors to consider is focusing more of the Introduction and Discussion simply on how the by-product effects of harvesting may be worsened during heat waves. Relative to the paper's current focus on ecosystem function and species interactions during heatwaves, I believe that a focus on harvesting and by-catch is a simpler description of the dynamic that was explored by the study. This would also allow the authors to make connections to other published research about by-catch impacts or fishing-induced mortality during heat waves.

- **Response:** Thanks to the reviewer for the suggestion here. While we agree that our experiment also aligns well with the fisheries angle proposed, we have elected to keep the scope of the paper focused on ecology rather than fisheries and bycatch. While relevant to the theme of fisheries bycatch, we feel that our findings are more ecologically relevant. In particular, the findings that heatwaves altered prey behaviour and mortality after being fished, altered predator and scavenger activity levels, and increased consumption rates of predators/scavengers on tossed-back sub-legal clams have ecological relevance and we prefer to stick to an ecological theme. We think that this is still a highly relevant theme for the paper, and feel that the revisions suggested by this and the other reviewer strengthen the evidence that our results provide for connections between climate change, fishing, and ecological interactions. The relevant text in the Introduction on Lines 60-72 with additional fisheries-relevant content now reads:

While studies suggest that heatwaves can have ecological effects, adequate in situ field studies in nearshore systems are rare²⁵. Such field studies are necessary to identify ecological mechanisms that may drive population- and community-level changes imposed by climate change, and to predict and adapt to these changes. Moreover, although some studies exist^{26,27}, research on the potential ecological effects of anthropogenic disturbances during heatwaves is limited. The act of fishing, for example, can result in ecological shifts by reducing prey abundances through the removal of targeted and non-targeted (bycatch)

species²⁸. While this knowledge has underpinned calls for ecosystem based fisheries management since the 1990s²⁸, the ways in which heatwaves might influence the ecosystem effects of fishing remains understudied for many fisheries. As such, an understanding of how fishing during heatwaves influences predator-prey dynamics and fishing mortality is paramount for developing effective ecosystem based fisheries management as the climate continues to change^{28,29}.

One globally-common nearshore fishery that has the potential to alter ecological function is clam digging. Clam digging during low tide can be destructive, as it...

2. Another aspect of the study that concerned me is that the enclosure treatment did not fully prevent predation. Consequently, the only instance that promoted distinguish between environmentally-induced mortality and predator-induced mortality was the one experiment during the heatwave (June) after 24 hours. At that point, approximately 50% of clam mortality was due to environmental stress and 50% was due to predation. But other than that first 24 hour period during June, it was not possible to discern how much of the mortality was due to environment, predation, and scavenging. And if a significant amount of the predation was actually scavenging, then the cause of mortality is primarily fishing x environmental stress, not predation, which would contrast the main focus of the paper (heatwave and harvesting alters predator-prey interaction).

- **Response:** The reviewer makes a good point that our predator exclusion plots did not fully prevent predation. Herein, we were able to discern when clams solely died from environmental stress, as dead clams still had tissue in the shell. But the reviewer is correct in that we are unable to fully distinguish between predation and scavenging for the heatwave trial at 48 hours. However, the high number of clams eaten during the heatwave and the increased predator/scavenger activity certainly indicate that the heatwave 1) altered predator/scavenger-prey activity levels; and 2) resulted in increased indirect fishing mortality. While the relative contributions of environment, predation, and scavenging are not discernable from our data, our data do suggest that most clams that were able to physiologically survive the heatwave would have succumb to predation (either by crabs or by mudsnails preying on physiologically stressed, but still alive, clams). Even if the clams were consumed by snails and crabs after death, they were consumed at far higher rates during the heatwave than any other time, which would have major ramifications for energy transfer within the system. As such, we retain that our results represent strong evidence for shifts in predator-prey dynamics, and ecological functioning, facilitated by clam digging. For this reason, we have kept an ecological focus in the manuscript, but have added additional fisheries context in the Intro as explained in the response to the previous comment. We have also revised the previous paragraph in the Discussion surrounding this aspect of the experiment (i.e., causes of mortality), and have altered the text throughout the manuscript, to avoid making the strict case that the heatwave increased predation, and predation caused the clam mortalities. Instead, the text now focuses on alterations in predator/scavenger and prey activity levels and increased fishing mortality, discussing the nuances with interpreting the causes of mortality, and placing this in the context of ecological consequences. This text on Lines 253-270 of the Discussion now reads:

Although effective at minimizing crab predation, our exclusion plots did not keep 100% of crabs out³⁰. As such, while it is clear from our results that fished sub-legal clam mortality

*increases during heatwaves, we are unable to determine the relative contributions of physiological death, predation, and scavenging in causing mortality. Based on observations at 24 hours after fishing during the heatwave, ~55% of the observed mortality was due to physiological (environmental) death, as only 45% of the dead clams were eaten at that time point. However, most of the uneaten dead clams at 24 h were found to be eaten 48 h later in the predator exclusion plots, suggesting that dead clams were heavily scavenged upon after physiological death. While crabs can certainly scavenge on dead clams, we also found increased activity of mudsnails (*Ilyanassa obsoleta*) in the mesocosm plots during the June heatwave trial. Mudsnails are known to both scavenge on dead molluscs^{42,43}, and actively predate on dying and unhealthy molluscs⁴⁴. Indeed, we directly observed congregations of mudsnails consuming tissues of clams in our June trial. Regardless of the ultimate source of clam mortality, our results clearly demonstrate that fishing during heatwaves can drive higher levels of predator/scavenger activity, reduce prey activity and predator avoidance behaviour, and increase the consumption of dead and dying prey through increases in predation and/or scavenging. This provides strong evidence for ecological shifts in predator-prey dynamics driven by human-climate interactions. Such ecological shifts will have ramifications for the flow of energy throughout the ecosystem, as observed in other systems^{7,45}. Relative abundances of different predators, scavengers, and prey within a given system will influence heatwave effects on energy flow.*

3. Finally, it is unclear whether the handling of the clams during the set-up of the experiment created more, less, or similar amounts of stress than the recreational and commercial clamming activities. To place the results of this study in context, it would be nice to have survey results of sub-legal size clam mortality on flats throughout the area during the heatwave and non-heatwave months. Or, alternatively, if the researchers could present some data on how the set-up of the experiment did not stress the clams during the heatwave.

- **Response:** This is a good comment and one that was also raised by Reviewer 1. We did not include adequate Discussion regarding differences in stress between our method and typical clam digging methods in our initial submission. It is important to note that stress during fishing was part of the experiment and is within the context of the paper, as we aimed to assess the effects of fishing during a heatwave on clam behaviour and survival. However, clam digging method is something that is difficult to control for because not all clam diggers use the same methods when fishing for clams, as there are no standardized management rules for how to handle/hold clams while fishing. As such, there is really no information on just how common a particular clam fishing method is. Nonetheless, based on conversations with some clam fishers over the past 5 years, we know that clam diggers most often fish for the maximum possible duration that they can during low tide (typically 5-6 hours), particularly in the Gulf of St. Lawrence because tide cycles are semi-diurnal with small tidal ranges (days where clam diggers can fish are limited throughout the year). Furthermore, while some do, not all clam diggers will attempt to minimize stress on their catch by placing clams in water, as many diggers simply place their catch in an empty bucket because filling it with water makes it heavy and would increase physical fatigue. Anecdotally, during one of the trials, we observed a recreational clam digger fish for clams for a longer duration than our fishing and tethering process, place all clams in a bucket without water, and measure and toss back the sub-legals once they were finished

digging. Such an approach during a heatwave would expose the fished clams to both desiccation and extreme air temperatures for 5-6 hours, likely maximizing physiological stress. As such, it is unlikely that our fishing methodology and the exposure of clams to air would invoke more stress on the clams than would be invoked by the practices of many clam diggers. Nonetheless, there may exist some optimal clam digging method(s) that could reduce stress on the animals when fishing during heatwaves or other acute environmental stresses. Further studies exploring the influence of different holding conditions when clam digging during periods of environmental stress on sub-legal burrowing and survival are thus warranted. While we do not provide any data in this regard, we have dedicated a full paragraph in the Discussion to explicitly address this aspect of the study on Lines 228-252, which reads:

It is important to note that while our experiment attempted to mimic typical clam digging methods in the region, some aspects of our experiment deviated from what clam diggers would do. Notably, when digging for clams, clam diggers would not tether washers to their sub-legal catch as we did in our experiment. Unfortunately, clam digging method is something that is difficult to control for because not all clam diggers use the same methods when fishing for clams, as there are no standardized management rules for how to handle/hold clams while fishing. For example, some clam diggers will put the clams in a bucket of water while other diggers don't add water, they just keep the clams in a dry bucket. If using water, some clam diggers would change out this water regularly as we did while others would not. Some clam diggers will measure and toss clams back on the fly while others will wait until they are finished fishing. While our methods do not replicate all of the ways in which clam diggers fish for clams, clam diggers most often fish for the maximum possible duration that they can during low tide (typically 5-6 hours), particularly in the Gulf of St. Lawrence because tide cycles are semi-diurnal with small tidal ranges (days where clam diggers can fish are limited throughout the year). Furthermore, not all clam diggers will attempt to minimize stress on their catch by placing clams in water, as many diggers simply place their catch in an empty bucket because filling it with water makes it heavy and would increase physical fatigue. Anecdotally, during one of the trials, we observed a recreational clam digger fish for clams for a longer duration than our fishing and tethering process, place all clams in a bucket without water, and measure and toss back the sub-legals once they were finished digging. Such an approach during a heatwave would expose the fished clams to both desiccation and extreme air temperatures, likely maximizing physiological stress. As such, it is unlikely that our fishing methodology and the exposure of clams to air would invoke more stress on the clams than would be invoked by the practices of many clam diggers. Nonetheless, there may exist some optimal clam digging method that could reduce stress on the animals when fishing during heatwaves or other acute environmental stresses. Further studies exploring the influence of different holding conditions when clam digging during periods of environmental stress on sub-legal burrowing and survival are thus warranted.

NOTE: For each response: **blue text** denotes the general response; specific text changes are in parentheses with **red text** denoting changed text, and **gray text** denoting retained text. Line numbers refer to CLEAN manuscript.

Reviewer #1 (Remarks to the Author):

The authors have substantially improved the quality of the manuscript, and I am pleased by how thoroughly the authors responded to my earlier comments. I only have a few minor revisions listed below.

Response: Thanks again to the reviewer for their constructive and helpful feedback. We incorporate all minor suggestions below.

Introduction

Lines 86-87: Consider rephrasing to “particularly if prey predator avoidance behaviours decrease²⁴”. While decreased activity levels in many bivalves would make them more vulnerable to predators, decreased activity levels in general does not always increase vulnerability. Many predator avoidance behaviors instead rely on decreasing activity level to avoid predation (e.g. crabs reducing activity in the presence of fish or freezing behavior in mammals).

Response: Rephrased, but slightly different to the reviewer’s specific suggestion (for readability). The text now reads: “...particularly if predator avoidance behaviours decrease in prey²⁴.” (L XXX-XXX)

Methods

Figure S4: Should add color scale to figure.

Response: Fixed.

NOTE: For each response: **blue text** denotes the general response; specific text changes are in parentheses with **red text** denoting changed text, and **gray text** denoting retained text. Line numbers refer to CLEAN manuscript.

Line 400-403: The sentence structure here is a little confusing with how the parenthetical was embedded. Were holding conditions also designed to minimize physiological stress associated with air exposure? Or is this something else that not all clam diggers do? If it's the latter, add it to the parenthetical. If it's the former, either move the parenthetical to the end of the sentence or change the second half of the sentence starting with "as well as" into its own new sentence (e.g. Holding conditions were also designed to minimize...).

Response: Rephrased accordingly and now reads, "Holding conditions aimed to mimic methods used by many clam diggers (based on personal conversations with a few clam diggers prior to experimentation; although not all clam diggers fill their buckets with water). Holding conditions were also designed to minimize physiological stress associated with air exposure..." (L 387-390)

Line 419: Here might be a good spot to mention that this time period was well within the amount of time diggers typically spend harvesting clams and then provide a normal range of time diggers are out harvesting. (Although it is mentioned in the discussion).

Response: Good suggestion. We've added a sentence to the end of the paragraph in the Methods as suggested, which now reads, "These fishing times are well within the daily time frames that clam diggers would be harvesting (typically 5-6 hours per day)." (L 405-407)

Line 503: "trails" should be "trials"

Response: Fixed.

Line 513: Listing the three categorical variables in the model as a single 3-way interaction in the model description seems a little unconventional. Did you only include the 3-way interaction or did you also include the lower order 2-way interactions? If you did include them, at least make "interactions" plural in the sentence.

Response: Good catch. The model tested for all independent and interactive effects of these three categorical variables (as can be seen in Table S6). We've amended the text accordingly, which now reads, "Models included average air temperature during fishing as a continuous smooth term, the independent and interactive effects of predator treatment, time since fishing, and tide level as categorical parametric terms, and Julian date as a continuous smooth random term." (L 506-508)

Lines 521-522: What is meant by "fishes the traps approximately"? Seems like an incomplete phrase on how frequently traps were fished.

*Response: We've amended the text for clarity, which now reads, "This annual program, designed to monitor invasive European green crabs (*Carcinus maenas*) in the park⁶⁷, deploys a series of modified eel traps at various sites throughout Kouchibouguac National Park. From June-October, park staff fish the traps every 1-2 weeks, recording counts of all crab species found in the traps..." (L 512-516)*

Figure S1 caption: The note about the May trial not being represented is stated twice.

Response: Fixed. We removed the first mention of this in parentheses after "...experiments conducted in 2024.". The supplementary figure caption now reads, "Figure S1. Average crab catches per unit effort at the 'Loggiacraft' monitoring site from the Kouchibouguac crab monitoring program from 2019-2024. Shaded areas in the top panels denote the timing and duration of each of the three-day mesocosm experiments conducted in 2024. Each point represents an average of 1-4 traps within a year (n = 4, 4, 1, 2, 2, 2 sequentially from 2019-2024). Note that the dates of the crab monitoring program did not capture the days of our May experiment."

Results

NOTE: For each response: **blue text** denotes the general response; specific text changes are in parentheses with **red text** denoting changed text, and **gray text** denoting retained text. Line numbers refer to CLEAN manuscript.

Line 115: “experimental trials” should be “experimental trials”.

Response: Fixed.

Line 116: change “The May experiment” to “The May trial”

Response: Text amended as suggested and now reads, “The May trial had the lowest mean air temperature...” (L 104-106)

Lines 184-185: Need to finish the phrase “which is consistent [with the BGLMM results]”.

Response: Text amended as suggested and now reads, “There were no significant effects of any other fixed factors on reburrowing, which is consistent with the BGLMM models...” (L 166-167)

Discussion

Line 254: rephrase to “do not add water, but just keep the clams in a dry bucket”

Response: Rephrased as suggested and now reads, “For example, some clam diggers will put the clams in a bucket of water, while other diggers do not add water, but just keep the clams in a dry bucket because filling it with water makes it heavy and would increase physical fatigue.” (L 232-234)

Lines 260-262: The information in this sentence is repetitive of the information provided in lines 252-254. Should integrate this information there.

Response: We have removed this sentence and have added the latter context (i.e., buckets with water are heavy) to the sentence above as suggested. We also slightly amended some text to increase flow and readability. The text here now reads, “For example, some clam diggers will put the clams in a bucket of water while other diggers do not add water, but just keep the clams in a dry bucket because filling it with water makes it heavy and would increase physical fatigue; if using water, some clam diggers would change out this water regularly as we did while others would not. Additionally, some clam diggers will measure and toss clams back on-the-fly while others will wait until they are finished fishing. While our methods do not replicate all of the ways in which clam diggers fish for clams, clam diggers most often fish for the maximum possible duration that they can during low tide (typically 5-6 hours; well beyond our fishing times), particularly in the southern Gulf of St. Lawrence because tide cycles are semi-diurnal with small tidal ranges (days where clam diggers can fish are limited throughout the year). Anecdotally, during one of the trials, we observed a recreational clam digger fish for clams for a longer duration than our fishing and tethering process, place all clams in a bucket without water, and measure and toss back the sub-legals once they were finished digging.” (L 232-243)

NOTE: For each response: **blue text** denotes the general response; specific text changes are in parentheses with **red text** denoting changed text, and **gray text** denoting retained text. Line numbers refer to CLEAN manuscript.

Reviewer #2 (Remarks to the Author):

I reviewed the original submission by Clements et al and have now reviewed the revised submission (COMMSBIO-24-6984A). I am concerned about the authors' inferences throughout the paper given the study design and results. In my initial read of the resubmission, I strongly disagreed with most of the author's conclusions about the role of predation because there was so much clam mortality in the predator exclosures (presumably due to physiological stress, not predation) and because the purported predator species (crab and snail) are well-known scavengers of dead things. Furthermore, while some predators may increase their foraging activity under elevated temperature, it's also well appreciated that more dead things lead to more scavenger activity. Consequently, a large abundance of dead clams could explain the higher activity of crabs and snails. At this point, I felt that the more conservative interpretation of the results was that heat and fishing interactively caused clam mortality, with a trophic benefit to scavengers.

But then I went for a walk and thought further about this paper. I revisited the results (specifically, Figure 2) and noted that the clam mortality after 24 hours in the predator exclosure treatment was ~ 35%. In contrast, the mortality of clams after 24 hours in the predator inclusion treatment was ~70%. Unfortunately, the predator exclosure treatment did not fully exclude all predators. But if we conservatively assume that all dead clams in the predator exclosure treatment were killed solely by heat and physiological stress, then I inferred that half of the mortality of clams in the predator inclusion treatment could be allocated to predation and the other ½ would be allocated to physiological stress (promoting scavenging). Furthermore, because the clam mortality after 48 hours was equally high in the predator exclosure and inclusion treatments, this enhanced predation interaction would be rather ephemeral, quickly converting from predation to scavenging as day 1 (24 hours) proceeded into day 2 (48 hours). This inference decreased my overall concern.

However, I then revisited the authors' rebuttal letter and the revised submission with a focus on paragraph five of the Discussion. The rationale and logical argument of this paragraph did not alleviate my concern. I suggest that the authors revise this paragraph to walk the reader through the situation I outlined above, which will highlight a potential role for predation would be equal in strength to stress-induced mortality, but of much shorter duration.

Response: Major thanks to the reviewer for the detail and logic outlined in this comment. We fully agree with the reviewer's suggestions here. However, rather than relying on eyeballed estimates of mortality rates in Figure 2, we elected to incorporate percentages estimates of mortality rates and the percentages of eaten clams in the different treatments and time points (as presented in Table 1) to generate a more complete picture of ecological dynamics for this system during the heatwave, in the context of predation and physiological death/scavenging . Our overall conclusion remains the same as the reviewer's, and we now make this explicit in the text (i.e., while predation is likely a large proximate cause of mortality at 24h, it is ephemeral and the majority of clams are likely to succumb to physiological death and be scavenged after 48 h even in the absence of predation). We have revised P5 of the Discussion accordingly, which reads:

“Although effective at minimizing crab predation, our exclusion plots did not keep 100% of crabs out³⁰. As such, while it is clear from our results that fished sub-legal clam mortality increases during heatwaves, we are unable to determine the relative contributions of physiological death, predation, and scavenging in causing mortality with a high degree of precision. Nonetheless, we can make some inferences based on observed mortality rates and the proportions of eaten vs. uneaten clams in the respective treatments (Table 1) to generate approximations. In the predator exclusion treatments, clam mortality after 24 h in the June trial was 24%, with 44% of

NOTE: For each response: **blue text** denotes the general response; specific text changes are in parentheses with **red text** denoting changed text, and **gray text** denoting retained text. Line numbers refer to CLEAN manuscript.

those clams ($\approx 10\%$ of all clams) found with tissues eaten. Unfortunately our predator exclusion cages did not keep 100% of predators out. However, conservatively assuming that all clams in the predator exclusion cages died physiologically, and that eaten clams were scavenged upon, suggests that, in the absence of predation, 14% of clams physiologically die and are not scavenged, 10% physiologically die are scavenged upon, and 76% survive. In contrast, 24 h June mortality in the predator inclusion treatments was 76%, with 72% of those dead clams (55% of total clams) being found with their tissues eaten. Interestingly, the 28% of dead clams (21% of all clams) that were uneaten in the predator inclusion treatments closely approximates the inferred 24% physiological mortality rate (with and without scavenging) in the predator exclusion treatments as above. This suggests that, when predators are present, scavenging is reduced at 24 h and the majority of eaten clams are likely consumed by predators; if scavenging was occurring, we would expect the portion of uneaten dead clams in the predator inclusion treatment to closer approximate the 14% of clams that physiologically died but were not scavenged in the predator exclusion treatment. As such, we can estimate that, when predators are present, $\approx 55\%$ of tossed back sub-legal clams are likely consumed by predators, $\approx 20\text{-}25\%$ succumb to physiological death, and $\approx 20\text{-}25\%$ survive 24 h after fishing during extreme heatwaves; however, when predators are largely absent, $\approx 20\text{-}25\%$ of clams succumb to physiological death after 24 h, with $\approx 45\%$ of those ($\approx 10\%$ of all clams) scavenged upon. Interestingly, at 48 h after fishing, the overall mortality rate of clams in the predator exclusion treatments increased tremendously, from 24% (at 24 h) to 81% (+53%), with 95% of those dead clams being eaten. Since predator exclusion treatments excluded most (but not all) predators, this suggests a high rate of physiological death and scavenging 48 h after being tossed back to the sediment. While crabs can certainly scavenge on dead clams, we also found increased activity of mudsnails (*Ilyanassa obsoleta*) in the mesocosm plots during the June heatwave trial, which are known to both scavenge on dead molluscs^{42,43}, and actively predate on dying and unhealthy molluscs⁴⁴. Indeed, we directly observed congregations of mudsnails consuming tissues of clams in our June trial. Similarly, mortality in the predator inclusion treatment increased from 76% at 24 h to 99% at 48 h (+23%). Given the drastic increase in mortality in the predator exclusion treatment at 48 h, coupled with the presence of both live and uneaten clams in predator inclusion treatments at 24 h, it is probable that this increase in mortality rate between 24 and 48 h in the predator inclusion treatments was also due to physiological death and scavenging, rather than predation. This ultimately indicates that while predation can serve as a proximate cause of mortality in tossed back sub-legal clams within 24 h, the role of predation appears ephemeral and, even in the absence of direct predation, the majority of clams are likely to succumb to physiological death and be scavenged after 48 h. Nonetheless, our results provide strong evidence for ecological shifts driven by human-climate interactions that are likely to have ramifications for the flow of energy throughout the ecosystem, as observed in other systems^{7,45}. Relative abundances of different predators, scavengers, and prey within a given system will influence heatwave effects on energy flow.” (L 250-290)

Another observation is that fourth paragraph of the Discussion about clam digging methods is nice, but it seems overkill. I suggest trimming it down.

Response: While we can appreciate that this paragraph may seem overkill, we want to note here that this article is likely to be read by scientists, resource managers, practitioners, and clam industry members from across the globe. As such, we feel it is worth highlighting various clam digging attributes in detail and how they relate to the results of our study. Furthermore, this paragraph was added at the request of another Reviewer, who wanted to see detailed discussion regarding the applicability of our method to actual clam fishing methods. We have thus elected

NOTE: For each response: **blue text** denotes the general response; specific text changes are in parentheses with **red text** denoting changed text, and **gray text** denoting retained text. Line numbers refer to CLEAN manuscript.

to largely keep this paragraph as it was in the previous revision, with the exception of removing a repetitive sentence (as suggested by Reviewer 1) and altering some of the text to enhance flow. We note that the word count of the main text of the article still falls well below the maximum suggested by the journal, and feel that this paragraph, as written, is useful for a broad suite of readers. The text here now reads, “It is important to note that while our experiment attempted to mimic typical clam digging methods in the region, some aspects of our experiment deviated from what clam diggers would do. Notably, when digging for clams, clam diggers would not tether washers to their sub-legal catch as we did in our experiment. Unfortunately, clam digging method is something that is difficult to control for because not all clam diggers use the same methods when fishing for clams, as there are no standardized management rules for how to handle/hold clams while fishing. For example, some clam diggers will put the clams in a bucket of water while other diggers **do not** add water, **but** just keep the clams in a dry bucket **because filling it with water makes it heavy and would increase physical fatigue; if** using water, some clam diggers would change out this water regularly as we did while others would not. **Additionally,** some clam diggers will measure and toss clams back **on-the-fly** while others will wait until they are finished fishing. While our methods do not replicate all of the ways in which clam diggers fish for clams, clam diggers most often fish for the maximum possible duration that they can during low tide (typically 5-6 hours; **well beyond our fishing times**), particularly in the **southern** Gulf of St. Lawrence because tide cycles are semi-diurnal with small tidal ranges (days where clam diggers can fish are limited throughout the year). Anecdotally, during one of the trials, we observed a recreational clam digger fish for clams for a longer duration than our fishing and tethering process, place all clams in a bucket without water, and measure and toss back the sub-legals once they were finished digging. Such an approach during a heatwave would expose the fished clams to both desiccation and extreme air temperatures, likely maximizing physiological stress. As such, it is unlikely that our fishing methodology would invoke more stress on the clams than would be invoked by the practices of many clam diggers. Nonetheless, there may exist some optimal clam digging method that could reduce stress on the animals when fishing during heatwaves or other acute environmental stresses. Further studies exploring the influence of different holding conditions when clam digging during periods of environmental stress on sub-legal burrowing and survival are thus warranted.” (L 227-249)

In addition, in paragraph 3 of the Discussion, the central argument is that increasing temperature enhances predator activity and that is what happened in the current study. But there is zero mention of a simpler explanation....heat waves x human disturbance can increase mortality of clams. Lots of dead clams attract more scavengers. At least revise this paragraph to give this simpler explanation an equal chance.

Response: While Paragraph 3 is not centered on predator activity exclusively, one of the arguments in this paragraph states that, “...low oxygen cannot explain the observed increase in predator activity during the June trial, which is most likely a consequence of higher temperatures.” We agree with the reviewer that attraction to dead and physiologically unhealthy clams can increase predator activity. We have altered the text in the paragraph to give equal weight to increased chemical attraction as suggested, noting that a combination of this and elevated temperature likely played a role. The text now reads, “However, low oxygen cannot explain the observed increase in predator activity during the June trial, which is most likely a consequence of higher temperatures **and/or the chemical attraction of predators by dead and dying clams**. Unfortunately, we did not measure water temperature or oxygen in the buckets directly. Finally, in our experiment, clams were exposed to air for a short duration while washers and fishing line were glued onto them. Air exposure during periods of high temperatures has

NOTE: For each response: *blue text* denotes the general response; specific text changes are in parentheses with *red text* denoting changed text, and *gray text* denoting retained text. Line numbers refer to CLEAN manuscript.

been reported to increase physiological stress and mortality in other bivalves^{40,41}. As such, exposure to high air temperatures during this time **likely** had negative effects on physiology and survival in the June trial. Ultimately, while we cannot pinpoint proximate causes of reduced burrowing and non-predation related mortality, it is very likely that high temperatures, low oxygen, and/or some combination of these stressors during fishing negatively affected clam physiology, leading to **poor physiological health**, increased vulnerability to predators, and contributing to the observed ecological shifts during the heatwave. Our results align with other recent findings that predator avoidance behaviour is reduced during heatwaves, which can alter ecological interactions between predators and their prey²⁴.” (L 214-226)

Finally, the abstract is well written, but it opens so generally that a reader could envision a central topic of study to be that humans are using their air conditioning during a heat wave or over using freshwater etc. I suggest revising the opening sentence so that it is a less vague.

Response: Fair point! We have altered the opening sentence of the Abstract (and subsequent sentences) to be less vague. It now reads, “Heatwaves may have multifaceted ecological impacts; however, field studies assessing the ecological ramifications of nearshore fishing during heatwaves are rare. We leverage a field experiment simulating clam fishing to document such effects on a ubiquitous ecological system at the land-sea interface. During monthly field trials from May-September 2024, we experimentally fished clams at low tide and tracked reburrowing and mortality rates of marked, sub-legal sized clams returned to the sediment.” (L 17-20)

Given this change in the abstract, we have also altered the title to be less vague with respect to ‘human activity’. Given the changes in the Discussion with respect to the specific role of predation, we also changed the term ‘predator-prey activity’ to ‘ecological interactions’ in the title as well: “Fishing during extreme heatwaves alters ecological interactions and increases indirect fishing mortality in a ubiquitous nearshore system”